# A Median Perspective on Unlabeled Data for Out-of-Distribution Detection

## Abstract

Out-of-distribution (OOD) detection plays a crucial role in ensuring the robustness and reliability of machine learning systems deployed in real-world applications. Recent approaches have explored the use of unlabeled data, showing potential for enhancing OOD detection capabilities. However, effectively utilizing unlabeled in-the-wild data remains challenging due to the mixed nature of both in-distribution (InD) and OOD samples. The lack of a distinct set of OOD samples complicates the task of training an optimal OOD classifier. In this work, we introduce Medix, a novel framework designed to identify potential outliers from unlabeled data using the median operation. We use the median because it provides a stable estimate of the central tendency, as an OOD detection mechanism, due to its robustness against noise and outliers. Using these identified outliers, along with labeled InD data, we train a robust OOD classifier. From a theoretical perspective, we derive error bounds that demonstrate Medix achieves a low error rate. Empirical results further substantiate our claims, as Medix outperforms existing methods across the board in open-world settings, confirming the validity of our theoretical insights.

## 1 Introduction

Deploying machine learning models in real-world applications often exposes them to challenges related to safety and reliability, particularly due to the presence of out-of-distribution (OOD) data. These OOD samples, stemming from unknown categories, should not be predicted by the model. However, neural networks are inherently vulnerable and typically lack the necessary mechanisms to detect and appropriately handle OOD inputs in practice (Nguyen et al., 2015).

Identifying OOD samples during inference is critical yet inherently not easy, as models are not exposed to unknown distributions during training and, therefore, cannot reliably distinguish OOD from in-distribution (InD) data. To address this challenge, recent approaches (Katz-Samuels et al., 2022a; Du et al., 2024a) have explored leveraging additional "in-the-wild" data to improve OOD detection. Specifically, Katz-Samuels et al. (2022a) introduced a method that uses unlabeled wild data for regularizing model training, while still focusing on classifying labeled InD data. The advantage of using such unlabeled wild data lies in its availability—being easily collectible once a model is deployed in its operating environment. This approach allows the model to better capture the true distribution of OOD data encountered during test time, leading to a more robust OOD detection.

However, leveraging unlabeled wild data presents significant challenges due to the complex mixture of InD and OOD data. The absence of a distinct and clean set of OOD samples complicates the development of robust OOD detection methods, especially since the OOD detector model only encounters data drawn from this mixed distribution, without knowledge of whether each sample is from the InD or OOD category. At present, the problem remains underexplored, with substantial opportunities for further advancement. Moreover, few studies establish a formal theoretical foundation, and to the best of our knowledge, Du et al. (2024a) is the only work that provides such a foundation for the "in-the-wild" setting.

Meanwhile, recent studies have demonstrated the effectiveness of median-based approaches in data pruning (Acharya et al., 2024). Prompted by these developments, this paper aims to answer the following question:

> How can median-based methods leverage unlabeled wild data to facilitate OOD detection with theoretical guarantees?

In an attempt to provide an affirmative answer to this question, we introduce a novel median-centric perspective for OOD detection. Specifically, we propose a median-based optimization framework and develop an algorithm, Medix, that effectively identifies OOD samples from wild data with low error rate. We provide a theoretical foundation that guarantees minimal error, which is further validated through experiments, demonstrating the robustness and efficiency of our algorithm. We are one of the few studies that provide such a theoretical foundation for the unlabeled "in-the-wild" setting. We show that median-based filtering is robust for outlier detection in unlabeled mixtures by bounding the fraction of InD samples incorrectly flagged as outliers. This fraction is controlled by two effects: the contamination effect, which quantifies the impact of OOD points and remains manageable as long as OOD proportion is below 50%; and the concentration effect, which leverages the sub-Gaussian nature of InD gradients to bound deviations of InD points from their mean gradient. Together, these effects ensure that, with high probability, the number of misclassified InD points remains small, demonstrating the method's effectiveness even in the worst-case scenario.

We benchmark our approach against two categories of methods: (1) those trained solely on InD data, and (2) those trained with both InD data and an auxiliary unlabeled dataset. On CIFAR-100 (Krizhevsky et al., 2009), Medix demonstrates a significant improvement over the strong baseline KNN+ (Sun et al., 2022), outperforming it by an average of 40.98% in terms of FPR95. Unlike approaches such as Outlier Exposure (Hendrycks et al., 2019), which rely on a clean, auxiliary unlabeled dataset (i.e. they make a strong distributional assumption that the auxiliary data is completely separable from the InD data), Medix achieves superior results without such assumptions, offering greater flexibility. Compared to WOODS (Katz-Samuels et al., 2022a), Medix reduces the average FPR95 by 1.32% on CIFAR-100 and 2.60% on CIFAR-10.

Our key contributions are as follows:

**C1)** We propose Medix, a median-centric greedy approach that filters outliers from the unlabeled wild data and then trains an OOD detector on the identified outliers and the InD samples. Our main contribution is the filtering stage.

**C2)** We establish theoretical guarantees for the robustness of median-based filtering in identifying both inliers and outliers within unlabeled mixtures. Specifically, we prove that the misclassification rates for both InD samples flagged as outliers and OOD samples retained as inliers are tightly controlled by two effects: the contamination effect, which remains bounded as long as the proportion of OOD samples is below 50%, and the concentration effect, which ensures stability in the gradient behavior of InD samples.

**C3)** We conduct an extensive evaluation of Medix across eleven InD-OOD pairs, comparing its performance against 20 competitive baselines. Our results demonstrate that Medix outperforms all the baselines, achieving superior performance across the board. We show via experiments that Medix outlier extraction achieves a low error rate (e.g. only 12.5%; see Figure 2), corroborating our theoretical findings.

## 2 PRELIMINARIES AND PROBLEM SETUP

In this section, we first provide an overview of the OOD detection problem, and then formally define the data setup, model architecture, loss functions, and the learning goal.

**Labeled In-Distribution Data.** Consider the input space $\mathcal{X}$ and the label space $\mathcal{Y} = \{1, \ldots, K\}$, which together define the structure of InD data. A labeled dataset $\mathcal{S}_{\mathrm{in}} = \{(\boldsymbol{x}_1, y_1), \ldots, (\boldsymbol{x}_n, y_n)\}$ is generated by sampling $n$ pairs independently and identically distributed (i.i.d.) from $\mathbb{P}_{XY}$, an unknown joint probability distribution over $\mathcal{X} \times \mathcal{Y}$. The marginal distribution of $\mathbb{P}_{XY}$ on $\mathcal{X}$ is denoted as $\mathbb{P}_{\mathrm{in}}$, representing the underlying distribution of InD inputs. We use $\mathcal{S}_{\mathrm{in}}$ to train an InD model.

**Out-of-distribution detection.** We address a practical scenario where the model is trained using labeled InD data but is later deployed in environments that may contain OOD inputs from classes not represented in the training data, i.e., for some label $y \notin \mathcal{Y}$. The model is expected to abstain from making predictions for such OOD inputs. At inference time, the primary objective is to determine whether a given input belongs to the InD distribution or arises from an OOD source.

**Unlabeled wild data.** One of the primary obstacles in OOD detection is the scarcity of labeled OOD samples. The potential sample space of OOD data can be very large, making the collection of labeled examples both costly and impractical. To address this, we introduce unlabeled wild data, $\mathcal{S}_{\text{wild}} = \{\tilde{\boldsymbol{x}}_1, \ldots, \tilde{\boldsymbol{x}}_m\}$, into our learning framework to better mimic real-world scenarios as proposed by Katz-Samuels et al. (2022a). Wild data is a blend of InD and OOD samples and can be readily collected during the deployment phase of a pre-trained model on $\mathcal{S}_{\text{in}}$. Similar to Du et al. (2024a); Katz-Samuels et al. (2022a), we adopt the Huber contamination model (Huber, 1964) to characterize the marginal distribution of the wild data

$$\mathbb{P}_{\text{wild}} := (1 - \pi)\mathbb{P}_{\text{in}} + \pi\mathbb{P}_{\text{out}}, \quad \pi \in (0, 1], \tag{1}$$

where $\pi$ denotes the contamination proportion and $\mathbb{P}_{\text{out}}$ captures the OOD distribution over $\mathcal{X}$. We note that the scenario where $\pi = 0$ corresponds to the absence of OOD samples, rendering the problem trivial.

**Models and Loss Functions.** Let $f_\phi : \mathcal{X} \to \mathbb{R}^K$ represent the InD classifier parameterized by $\phi \in \Phi$, where $\Phi$ denotes the parameter space for this classifier. The output of $f_\phi$ corresponds to a soft probability distribution over the $K = |\mathcal{Y}|$ InD classes. The loss function for the labeled InD data is defined as $\ell : \mathbb{R}^K \times \mathcal{Y} \to \mathbb{R}$. For OOD detection, we introduce a separate classifier $g_\theta : \mathcal{X} \to \mathbb{R}$, parameterized by $\theta \in \Theta$, with $\Theta$ as the parameter space. The binary loss function associated with $g_\theta$ is denoted as $\ell_b(g_\theta(x), y_b)$, where $y_b \in \mathcal{Y}_b := \{y_+, y_-\}$. Here, $y_+ > 0$ represents the InD class, while $y_- < 0$ corresponds to the OOD class.

**Learning objective.** Our learning framework is designed to simultaneously train the OOD detector $g_\theta$ and the multi-class classifier $f_\phi$, leveraging both the InD data $\mathbb{P}_{\text{in}}$ and the wild data $\mathbb{P}_{\text{wild}}$. During testing, we evaluate the performance using the following metrics:

$$\downarrow \text{FPR}(g_\theta) = \mathbb{E}_{\boldsymbol{x} \sim \mathbb{P}_{\text{out}}^{\text{test}}}\left[\mathbb{I}\{g_\theta(\boldsymbol{x}) = \text{in}\}\right],$$
$$\uparrow \text{TPR}(g_\theta) = \mathbb{E}_{\boldsymbol{x} \sim \mathbb{P}_{\text{in}}}\left[\mathbb{I}\{g_\theta(\boldsymbol{x}) = \text{in}\}\right],$$
$$\uparrow \text{Acc}(f_\phi) = \mathbb{E}_{(\boldsymbol{x},y) \sim \mathbb{P}_{XY}}\left[\mathbb{I}\{f_\phi(\boldsymbol{x}) = y\}\right],$$

where $\mathbb{I}\{\cdot\}$ denotes the indicator function, and $\mathbb{P}_{\text{out}}^{\text{test}}$ represents the OOD test data distribution.

## 3 METHOD: MEDIAN-CENTRIC FRAMEWORK FOR OOD DETECTION

In this section, we present a novel learning paradigm, termed **Medix**, designed for OOD detection by harnessing the power of unlabeled wild data. Our framework overcomes the limitations of conventional approaches that rely exclusively on InD data and is particularly well-suited for applications in open-world environments, where models are often confronted with previously unseen inputs. The Medix framework is composed of two integral stages: **1) Outlier Extraction**: A filtering process that isolates candidate OOD samples from the unlabeled wild data (explained in Section 3.1), and **2) Detector Training**: train a binary OOD detector using both InD data and the outlier candidates identified in the previous step (explained in Section 3.2). For stage 2, we follow the protocol introduced by Du et al. (2024a). As we will demonstrate in the subsequent sections, this two-step methodology not only facilitates the effective extraction of OOD data from the unlabelled wild data but also establishes a robust foundation for deploying machine learning models in dynamic, open-world scenarios.

### 3.1 EXTRACTING CANDIDATE OUTLIERS FROM THE WILD DATA

To isolate potential outliers from the wild mixture $\mathcal{S}_{\text{wild}}$, our framework leverages an optimization-based approach that exploits the gradients of the model parameters. These gradients are derived from a classification model, $f_\phi$, which is trained solely on the InD dataset $\mathcal{S}_{\text{in}}$. The detailed methodology for this process is formally outlined below.

**Reference gradient estimation from InD data.** The first step in our proposed framework is to estimate a reference gradient using the InD dataset $\mathcal{S}_{\text{in}}$. This is achieved by training a classifier $f_\phi$ on $\mathcal{S}_{\text{in}}$ through empirical risk minimization (ERM) as follows

$$\phi_{\mathcal{S}_{\text{in}}} \in \arg\min_{\phi \in \Phi} \mathcal{L}_{\mathcal{S}_{\text{in}}}(f_\phi), \quad \text{where} \quad \mathcal{L}_{\mathcal{S}_{\text{in}}}(f_\phi) = \frac{1}{n}\sum_{(\boldsymbol{x}_i, y_i) \in \mathcal{S}_{\text{in}}} \ell(f_\phi(\boldsymbol{x}_i), y_i), \tag{2}$$

where $\phi_{\mathcal{S}_{\text{in}}}$ denotes the learned parameters. Once the classifier has been trained, we compute the mean gradient $\bar{\nabla}_{\text{in}}$ as the average of the gradients of the loss function with respect to the model parameters

over the InD data:

$$\bar{\nabla}_{\text{in}} = \frac{1}{n} \sum_{(\boldsymbol{x}_i, y_i) \in \mathcal{S}_{\text{in}}} \nabla\ell(f_{\phi_{\mathcal{S}_{\text{in}}}}(\boldsymbol{x}_i), y_i). \tag{3}$$

In our approach, $\bar{\nabla}_{\text{in}}$ serves as the reference gradient, enabling the quantification of deviations for other data points relative to this reference.

**Motivation.** We hypothesize that increasing the number of OOD samples in the wild dataset, $\mathcal{S}_{\text{wild}}$, will lead to a greater deviation from the average InD gradient, $\bar{\nabla}_{\text{in}}$. To test this hypothesis, we design an initial experiment using CIFAR-10 (Krizhevsky et al., 2009) as the InD dataset and SVHN (Netzer et al., 2011) as the OOD dataset. Specifically, $\mathcal{S}_{\text{wild}}$ consists of 10,000 samples drawn from CIFAR-10, ensuring that these samples are disjoint from the training set used to train the model $\phi_{\mathcal{S}_{\text{in}}}$, which we leverage to compute $\bar{\nabla}_{\text{in}}$. We incrementally add SVHN OOD samples to $\mathcal{S}_{\text{wild}}$ and track the behavior of the $L_2$-norm deviation between $\bar{\nabla}_{\text{in}}$ and the element-wise median (EWM) of the gradients of the wild dataset as follows:

$$\left\| \bar{\nabla}_{\text{in}} - \text{EWM}\left( \left\{ \nabla\ell\left(f_{\phi_{\mathcal{S}_{\text{in}}}}(\tilde{\boldsymbol{x}}_i), \hat{y}_{\tilde{\boldsymbol{x}}_i}\right) \right\}_{i \in \mathcal{S}_{\text{wild}}} \right) \right\|.$$

Our results, depicted in Figure 1, reveal a clear and monotonic increase in the $L_2$-norm deviation, supporting our hypothesis. This observation serves as a key motivation for the method we introduce in the subsequent section. Notably, the stopping criterion for our algorithm is derived from this monotonically increasing behavior, where we terminate the algorithm when the $L_2$-norm deviation between consecutive iterations drops below a threshold $\epsilon$; we will explain this method in detail in the following section.

**Filtering potential outliers from unlabeled wild data.** Motivated by the results in Figure 1, we formulate the following optimization problem to identify the outlier subset $\mathcal{S}_{\text{out}}^*$ in $\mathcal{S}_{\text{wild}}$:

$$\mathcal{S}_{\text{in}}^* = \arg\min_{\mathcal{S} \subseteq \mathcal{S}_{\text{wild}}} \left\| \bar{\nabla}_{\text{in}} - \text{EWM}(G_{\mathcal{S}}) \right\|, \quad \text{where} \quad G_{\mathcal{S}} = \left\{ \nabla\ell\left(f_{\phi_{\mathcal{S}_{\text{in}}}}(\tilde{\boldsymbol{x}}_i), \hat{y}_{\tilde{\boldsymbol{x}}_i}\right) \right\}_{i \in \mathcal{S}}. \tag{4}$$

Here $\text{EWM}(\cdot)$ denotes the element-wise median function, and $\hat{y}_{\tilde{\boldsymbol{x}}_i}$ represents the predicted label for a wild sample $\tilde{\boldsymbol{x}}_i$. For notational simplicity, we define $G_{\mathcal{S}} = \{\nabla\ell(\tilde{\boldsymbol{x}}_i)\}_{i \in \mathcal{S}}$.

The above optimization problem aims to identify a subset $\mathcal{S}$ in $\mathcal{S}_{\text{wild}}$ that minimizes the distance between the EWM of the gradients and the average gradient $\bar{\nabla}_{\text{in}}$. According to Figure 1, such a subset, denoted by $\mathcal{S}_{\text{in}}^*$, may well represent the InD data in $\mathcal{S}_{\text{wild}}$, in which case $\mathcal{S}_{\text{out}}^* = \mathcal{S}_{\text{wild}} \backslash \mathcal{S}_{\text{in}}^*$ may capture the OOD data in $\mathcal{S}_{\text{wild}}$.

Solving the optimization problem in equation 4 can be computationally prohibitive, especially as the size of $\mathcal{S}_{\text{wild}}$ increases. To address this, we propose a greedy approximation based on a leave-one-out approach, as outlined in Algorithm 1. The algorithm implements an iterative procedure for outlier

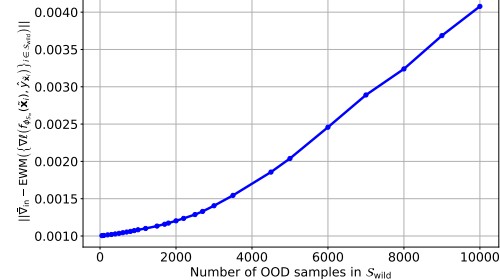

Figure 1: Distance deviation as we increase OOD samples in $\mathcal{S}_{\text{wild}}$.

detection from a wild dataset, leveraging deviations in gradient information relative to the InD dataset.

The algorithm begins by computing the EWM of the wild data gradients at each iteration, which serves as a reference for comparing against the average gradient of the InD data $\bar{\nabla}_{\text{in}}$. We denote by $d_t$, the $L_2$ distance between the average InD gradient $\bar{\nabla}_{\text{in}}$ and the EWM gradients of the data left in the wild set, represented by $\mathcal{S}$, i.e., $d_t = \|\text{EWM}(G_{\mathcal{S}}) - \bar{\nabla}_{\text{in}}\|$. The algorithm then iteratively identifies samples in $\mathcal{S}$ that incur the most significant drop in the $L_2$ distance with $d_t$ when removed from $\mathcal{S}$ as OOD. Specifically, having data samples $\mathcal{S}$ left, we remove each sample $i \in \mathcal{S}$ and compute the EWM as $\text{EWM}(G_{\mathcal{S} \backslash \{i\}})$ and the distance to $\bar{\nabla}_{\text{in}}$ as $\|\text{EWM}(G_{\mathcal{S} \backslash \{i\}}) - \bar{\nabla}_{\text{in}}\|$. We then find the drop in distance $\delta_i = d_t - \|\text{EWM}(G_{\mathcal{S} \backslash \{i\}}) - \bar{\nabla}_{\text{in}}\|$ and find the $k$ samples with the largest drop and identify them as OOD. The algorithm repeats until there is no significant drop in $\delta_i$ or a maximum number of iterations is reached. The convergence criterion is based on the change in the $L_2$ distance between two iterations, which must fall below a predefined $\epsilon$ threshold; this criterion is inspired by the monotonically increasing trend that we observed in our preliminary experiment in Figure 1, i.e.,

as a substantial number of OOD samples are removed, the $L_2$ distance gradually decreases to a small value, signaling the point at which the algorithm should halt to avoid erroneously identifying InD samples as outliers, thus preventing any degradation in performance.

---

**Algorithm 1** Iterative Outlier Detection via **Medix**

---

**Require:** $\bar{\nabla}_{\text{in}}$, $G_i = \nabla \ell(f_{\phi_{\mathcal{S}_{\text{in}}}}(\tilde{\boldsymbol{x}}_i), \hat{y}_{\tilde{\boldsymbol{x}}_i})$, maximum iterations $T$, hyperparameters $\epsilon$, $k$

1: Initialize $\mathcal{S} \leftarrow \mathcal{S}_{\text{wild}}$ (wild set), $\mathcal{O} \leftarrow \emptyset$ (outliers), $t \leftarrow 0$ (iteration), $d_t \leftarrow 0$, $\delta_{\max} \leftarrow \infty$ (deviations), $\mathcal{I}_k \leftarrow \emptyset$
2: **while** $t \leq T$ or $|\delta_{\max}| > \epsilon$ **do**
3:      $\mathcal{O} \leftarrow \mathcal{O} \cup \{\tilde{\boldsymbol{x}}_i : i \in \mathcal{I}_k\}$                ▷ Add outliers to set $\mathcal{O}$
4:      $d_t \leftarrow \|\text{EWM}(G_{\mathcal{S}}) - \bar{\nabla}_{\text{in}}\|$           ▷ Compute L2 deviation
5:      **for** each $i \in \mathcal{S}$ **do**
6:          $\delta_i \leftarrow d_t - \|\text{EWM}(G_{\mathcal{S} \setminus \{i\}}) - \bar{\nabla}_{\text{in}}\|$
7:      **end for**
8:      $\mathcal{I}_k \leftarrow \text{indices}(\text{top-}k(\{\delta_i\}_{i \in \mathcal{S}}))$       ▷ Select top-$k$ indices
9:      $\mathcal{S} \leftarrow \mathcal{S} \setminus \mathcal{I}_k$                  ▷ Remove outliers from $\mathcal{S}$
10:     $\delta_{\max} \leftarrow \max_{i \in \mathcal{S}}\{\delta_i\}$, $t \leftarrow t + 1$
11: **end while**
12: **return** $\hat{\mathcal{S}}_{\text{out}} = \mathcal{O}$                    ▷ Return detected outliers

---

### 3.2 Training the OOD Detector with Candidate Outliers

After identifying the candidate outlier set $\hat{\mathcal{S}}_{\text{out}}$ from the wild data, we proceed to train the OOD detector $g_\theta$ designed to maximize the distinction between InD and candidate outlier data following the protocol in Du et al. (2024a). The objective function explicitly enforces separability at the decision boundary (thresholded at 0), assigning positive outputs for labeled InD samples $\boldsymbol{x} \in \mathcal{S}_{\text{in}}$ and negative outputs for candidate outliers $\tilde{\boldsymbol{x}} \in \hat{\mathcal{S}}_{\text{out}}$. Specifically, the loss function is defined as:

$$\mathcal{L}_{\mathcal{S}_{\text{in}}, \hat{\mathcal{S}}_{\text{out}}}(g_\theta) = \mathcal{L}^+_{\mathcal{S}_{\text{in}}}(g_\theta) + \mathcal{L}^-_{\hat{\mathcal{S}}_{\text{out}}}(g_\theta),$$

$$\mathcal{L}^+_{\mathcal{S}_{\text{in}}}(g_\theta) = \mathbb{E}_{\boldsymbol{x} \in \mathcal{S}_{\text{in}}} \mathbb{I}\{g_\theta(\boldsymbol{x}) \leq 0\},$$

$$\mathcal{L}^-_{\hat{\mathcal{S}}_{\text{out}}}(g_\theta) = \mathbb{E}_{\tilde{\boldsymbol{x}} \in \hat{\mathcal{S}}_{\text{out}}} \mathbb{I}\{g_\theta(\tilde{\boldsymbol{x}}) > 0\}. \tag{5}$$

To address the challenge posed by the non-differentiable nature of the 0/1 loss, a binary loss based on a differentiable sigmoid function is employed as a smooth surrogate, ensuring tractable optimization while retaining alignment with the original objective. The learning framework incorporates the InD risk as defined in Equation 2, thereby safeguarding the predictive accuracy for InD samples. This unified optimization approach significantly bolsters the generalization performance of $g_\theta$, enabling robust detection of OOD samples drawn from $\mathbb{P}_{\text{out}}$, as corroborated by our results.

## 4 Theoretical Analysis

We now present the theoretical guarantees of Medix's filtering stage. The following theorems provide provable upper bounds on the misclassification rates for both InD and OOD points. Together, they demonstrate the two-sided robustness of our EWM filtering. For detailed proofs, see Appendix C.

> **Theorem 4.1** (Inlier Misclassification Bound). *Assume that the gradients of InD points in $\mathcal{S}_{\text{wild}}$ are i.i.d., and each coordinate is sub-Gaussian with variance proxy $\sigma^2$. Let $\epsilon = \sigma\sqrt{2\log(2dm_{\text{in}})}$, and fix any confidence level $\delta \in (0, 1)$. Then, with probability at least $1 - \delta$, the inlier misclassification rate of the EWM filtering rule satisfies:*
>
> $$\text{ERR}_{\text{in}} \leq \underbrace{\frac{1}{m_{in}} + 2\sqrt{\frac{\log(1/\delta)}{2m_{\text{in}}}}}_{\text{Concentration term}} + \underbrace{\frac{\pi}{2(1 - \pi)}}_{\text{Contamination term}}.$$

Theorem 4.1 reveals that the inlier misclassification rate is governed by two key effects: a *concentration term*, which vanishes with larger sample size and captures statistical fluctuations in the InD gradients, and a *contamination term*, which quantifies the influence of OOD points on the EWM. Notably, the bound remains controlled as long as the contamination ratio $\pi < 0.5$, underscoring the robustness of median-based filtering to moderate OOD presence. We now turn to the complementary question: *how many true outliers fail to be flagged and are mistakenly retained?* The next theorem answers this by bounding the OOD misclassification rate.

**Theorem 4.2** (Outlier Misclassification Bound). *Assume that the gradients of OOD points in* $\mathcal{S}_{\text{wild}}$ *are i.i.d., and each coordinate is sub-Gaussian with variance proxy* $\sigma_{\text{out}}^2$. *Suppose the mean OOD gradient* $\mu_{\text{out}}$ *satisfies a separation condition:*

$$\|\mu_{\text{out}} - \bar{\nabla}_{\text{in}}\|_2 \geq \Delta\sqrt{d},$$

*for some* $\Delta > 0$. *Then, for any tolerance* $\epsilon \in (0, \Delta)$ *and confidence level* $\delta \in (0, 1)$, *with probability at least* $1 - \delta$, *the outlier misclassification rate satisfies:*

$$\mathrm{ERR}_{\text{out}} \leq \underbrace{2d\exp\left(-\frac{(\Delta - \epsilon)^2}{2\sigma_{\text{out}}^2}\right)}_{\textit{Separation term}} + \underbrace{\sqrt{\frac{\log(1/\delta)}{2m_{\text{out}}}}}_{\textit{Concentration term}} + \underbrace{\frac{1 - \pi}{2\pi}}_{\textit{Contamination term}} \quad .$$

Theorem 4.2 complements the previous result by bounding the fraction of OOD points mistakenly retained as InD. Together, Theorems 4.1 and 4.2 provide a two-sided guarantee for the robustness of the Medix filtering method. The inlier and outlier misclassification rates are governed by a balance between the following three effects:

**Contamination Effect.** For Theorem 4.1, even when the wild dataset contains a significant fraction $\pi$ of OOD points, the median-based criterion remains stable as long as $\pi < 0.5$. This is reflected in the $\pi/[2(1 - \pi)]$ term of the bound, which increases with $\pi$ and encodes the risk that OOD gradients may skew the EWM. Conversely, Theorem 4.2 includes a symmetric penalty $\frac{1-\pi}{2\pi}$, quantifying the difficulty of isolating OOD points when they are underrepresented.

**Concentration Effect.** Both bounds exploit the sub-Gaussian nature of the gradient coordinates. For InD data, this ensures that most gradients concentrate near $\bar{\nabla}_{\text{in}}$, keeping the median stable even in finite samples. For OOD points, concentration around $\mu_{\text{out}}$ allows us to quantify the risk that a misaligned OOD sample slips past the filter. These concentration effects decay as $1/\sqrt{m}$, where $m$ is the number of samples from each class.

**Separation Effect.** Unique to Theorem 4.2, this effect quantifies how far the OOD mean gradient must lie from the InD mean gradient in order to reliably reject OOD samples. The exponential term $\exp(-(\Delta - \varepsilon)^2/2\sigma_{\text{out}}^2)$ captures this trade-off: the more separated the distributions, the less likely it is for OOD gradients to fool the filter.

> Theorems 4.1 and 4.2 jointly establish that, with high probability, median filtering achieves robust separation of InD and OOD samples in unlabeled data mixtures. The fraction of InD samples misclassified as outliers is bounded by contamination and concentration effects, while the fraction of OOD points incorrectly retained is governed by an exponential separation term, a concentration bound, and a reverse contamination effect. These results provide rigorous theoretical assurance that Medix minimizes both types of errors under mild assumptions.

*Remark* 4.3. *Sub-Gaussian Assumption for Gradient Coordinates.* The assumption that gradients of InD samples in each coordinate are sub-Gaussian is crucial for deriving concentration bounds and ensuring robustness in statistical estimation. Further empirical evidence as shown in Figure 4a confirms this assumption, where we observe that the histogram of gradient values for InD samples is bell-shaped and concentrated around the mean. This indicates that the gradients exhibit light tails, a defining characteristic of sub-Gaussian random variables, in which variables have exponentially decaying tails, meaning extreme gradient values are rare. This aligns with the observed behavior in the histogram. Furthermore, Figure 4b compares the empirical quantiles of the gradient distribution with the theoretical quantiles of a Gaussian distribution in a Q-Q plot (Wilk & Gnanadesikan, 1968). The close alignment of points with the 45-degree reference line demonstrates that the empirical distribution of gradients indeed closely resembles a Gaussian distribution. Since Gaussian random variables are a subset of sub-Gaussian variables, this supports the sub-Gaussian assumption. *We also provide a looser version of these bounds, which **removes the sub-Gaussian assumption** and holds under merely bounded second moments. This version is presented in Theorem C.3 (Appendix C.3). While the rates degrade, the core robustness guarantee of Medix still holds in this broader setting.*

## 5 EXPERIMENTS

This section will demonstrate the efficacy of Medix across various InD-OOD dataset pairs, benchmarking it against 20 widely-used baselines. All experiments are performed on hardware equipped with NVIDIA A100-SXM4-80GB GPUs. We provide the necessary code to reproduce our results.

### 5.1 MODELS, DATASETS, AND BASELINES

**Datasets.** We use the same experimental protocol as Katz-Samuels et al. (2022a), which introduced the problem of learning OOD detectors with wild data, enabling a fair comparison to prior work. Specifically, we use CIFAR-10 and CIFAR-100 as InD datasets ($\mathbb{P}_{in}$). For OOD testing, we select a suite of natural image datasets, including PLACES365 (Zhou et al., 2017), SVHN, TEXTURES (Cimpoi et al., 2014), and LSUN-RESIZE & LSUN-C (Yu et al., 2015) as OOD datasets ($\mathbb{P}_{out}$). To simulate wild data ($\mathbb{P}_{wild}$), we combine a subset of InD data ($\mathbb{P}_{in}$) with the OOD data ($\mathbb{P}_{out}$) under a default mixing parameter $\pi = 0.5$. For example, when using PLACES365 as an OOD test set, we construct a wild mixture by combining CIFAR with PLACES365 as wild data and test on PLACES365 as the OOD set. This procedure is repeated across all OOD datasets and baselines. The InD CIFAR dataset is split into two halves: the first 25,000 images to train $\phi_{\mathcal{S}_{in}}$, while the remaining images to generate the wild mixture $\mathcal{S}_{wild}$. For gradient computation, we use the penultimate layer weights, as these have been shown to be particularly informative for OOD detection (Huang et al., 2021a).

**Evaluation metrics.** We evaluate using three standard metrics: (1) False Positive Rate (FPR95↓) of OOD samples when the True Positive Rate of InD samples is 95%, (2) Area Under the Receiver Operating Characteristic Curve (AUROC↑), and (3) InD Classification Accuracy (InD Acc↑).

**Baselines.** Our comparison encompasses a diverse set of competitive baselines, categorized based on whether they are trained using only InD data or both InD and wild data. For the methods trained exclusively on InD data ($\mathbb{P}_{in}$), we compare Medix against a variety of established OOD detection methods, including Maximum Softmax Probability (MSP) (Hendrycks & Gimpel, 2016), ODIN (Liang et al., 2018), Mahalanobis Distance (Lee et al., 2018b), Energy Score (Liu et al., 2020c), ReAct (Sun et al., 2021), DICE (Sun & Li, 2022), KNN Distance (Sun et al., 2022), and ASH (Djurisic et al., 2022); these methods use a model trained with softmax cross-entropy loss. Additionally, we also compare against methods based on contrastive loss, such as CSI (Tack et al., 2020b) and KNN+ (Sun et al., 2022), for a more comprehensive comparison. For methods that leverage both InD and wild data, we compare against Outlier Exposure (OE) (Hendrycks et al., 2019) and energy-regularization learning (Liu et al., 2020c), which regularize the training by promoting lower confidence or higher energy on outlier data. We also include a comparison with WOODS (Katz-Samuels et al., 2022a), which introduced the concept of wild unlabeled data and utilizes it for OOD detection through a constrained optimization approach. Finally, we included more recent baselines, including CONJ (Peng et al., 2024) and DRL (Zhang et al., 2024), to provide a more thorough evaluation.

### 5.2 EXPERIMENTAL SETUP

In line with WOODS (Katz-Samuels et al., 2022a), we employ a Wide ResNet architecture (Zagoruyko, 2016) with 40 layers and a width factor of 2 for the InD classifier $\phi_{\mathcal{S}_{in}}$. It is trained using stochastic gradient descent with a momentum of 0.9, weight decay of 0.0005, and an initial learning rate of 0.1. Training is performed for 100 epochs with cosine learning rate decay, a batch size of 128, and a dropout rate of 0.3. Hyperparameters $\epsilon$ and $k$ used in the proposed method Medix are selected from the sets {5e-5, 5e-4, 5e-3, 5e-2} and {4k, 7k, 10k, 20k}, respectively, taking into account dataset sizes and with the objective of maximizing OOD performance. For the OOD classifier $g_\theta$, we initialize it with the pre-trained InD classifier $\phi_{\mathcal{S}_{in}}$ and add a linear layer that performs binary classification using the penultimate-layer features. The learning rate for this classifier is set to 0.001, and fine-tuning is done for 100 epochs as outlined in Equation equation 5. We combine the binary classification loss with the InD classification loss, assigning a weight of 10 to the binary classification component. All other training parameters remain the same as those used for training $\phi_{\mathcal{S}_{in}}$.

### 5.3 RESULTS

We present our main results in Table 2 on CIFAR-100, where Medix remarkably outperforms all OOD detection baselines. The results highlight the following key observations: (1) Methods trained on both InD and wild data significantly outperform those trained exclusively on InD data. Medix reduces the FPR95 by 52.31% on PLACES365 and 38.24% on TEXTURES compared to KNN+, demonstrating the effectiveness of incorporating in-the-wild data for model regularization. (2) Medix further outperforms competitive methods utilizing wild data ($\mathbb{P}_{wild}$): On CIFAR-100, Medix achieves an average FPR95 of 5.42%, which represents a 1.32% improvement over WOODS. Additionally, Medix maintains a competitive InD accuracy of 73.33%. This slight difference can be attributed to the fact that our method is trained on 25,000 labeled InD samples, while baseline methods, which do not leverage wild data, use the full CIFAR-100 training set of 50,000 samples. We present the results for CIFAR-10 in Table 1, where Medix surpasses all baseline methods. Medix outperforms

Table 1: OOD detection performance comparison of Medix and baselines on CIFAR-10 as InD data. Performance averaged over five runs; best results are highlighted in **bold**.

| Methods | SVHN | | PLACES365 | | LSUN-C | | LSUN-RESIZE | | TEXTURES | | Average | | InD ACC↑ |
|---|---|---|---|---|---|---|---|---|---|---|---|---|---|
| | FPR95↓ | AUROC↑ | FPR95↓ | AUROC↑ | FPR95↓ | AUROC↑ | FPR95↓ | AUROC↑ | FPR95↓ | AUROC↑ | FPR95↓ | AUROC↑ | |
| Using $\mathbb{P}_{in}$ only | | | | | | | | | | | | | |
| MSP | 48.49 | 91.89 | 59.48 | 88.20 | 30.80 | 95.65 | 52.15 | 91.37 | 59.28 | 88.50 | 50.04 | 91.12 | 94.84 |
| ODIN | 33.35 | 91.96 | 57.40 | 84.49 | 15.52 | 97.04 | 26.62 | 94.57 | 49.12 | 84.97 | 36.40 | 90.61 | 94.84 |
| Mahalanobis | 12.89 | 97.62 | 68.57 | 84.61 | 39.22 | 94.15 | 42.62 | 93.23 | 15.00 | 97.33 | 35.66 | 93.34 | 94.84 |
| Energy | 35.59 | 90.96 | 40.14 | 89.89 | 8.26 | 98.35 | 27.58 | 94.24 | 52.79 | 85.22 | 32.87 | 91.73 | 94.84 |
| KNN | 24.53 | 95.96 | 25.29 | 95.69 | 25.55 | 95.26 | 27.57 | 94.71 | 50.90 | 89.14 | 30.77 | 94.15 | 94.84 |
| ReAct | 40.76 | 89.57 | 41.44 | 90.44 | 14.38 | 97.21 | 33.63 | 93.58 | 53.63 | 86.59 | 36.77 | 91.48 | 94.84 |
| DICE | 35.44 | 89.65 | 46.83 | 86.69 | 6.32 | 98.68 | 28.93 | 93.56 | 53.62 | 82.20 | 34.23 | 90.16 | 94.84 |
| ASH | 6.51 | 98.65 | 48.45 | 88.34 | 0.90 | 99.73 | 4.96 | 98.92 | 24.34 | 95.09 | 17.03 | 96.15 | 94.84 |
| CSI | 17.30 | 97.40 | 34.95 | 93.64 | 1.95 | 99.55 | 12.15 | 98.01 | 20.45 | 95.93 | 17.36 | 96.91 | 94.17 |
| KNN+ | 2.99 | 99.41 | 24.69 | 94.84 | 2.95 | 99.39 | 11.22 | 97.98 | 9.65 | 98.37 | 10.30 | 97.99 | 93.19 |
| Using $\mathbb{P}_{in}$ and $\mathbb{P}_{wild}$ | | | | | | | | | | | | | |
| OE | 1.13 | 99.53 | 19.48 | 94.88 | 1.91 | 98.16 | 0.54 | 98.84 | 7.75 | 98.56 | 6.16 | 97.99 | 94.12 |
| Energy (w/ OE) | 5.24 | 99.41 | 14.66 | 96.18 | 2.35 | 99.30 | 4.85 | 98.62 | 10.51 | 97.10 | 7.52 | 97.98 | 94.24 |
| WOODS | 0.17 | 99.91 | 10.19 | 98.05 | 0.31 | 99.14 | 0.11 | 99.38 | 6.21 | 98.13 | 3.40 | 98.92 | 94.74 |
| Medix | **0.06** | **99.98** | **2.98** | **99.10** | **0.01** | **99.98** | **0.01** | **99.98** | **0.96** | **99.66** | **0.80** | **99.74** | 93.58 |
| (Ours) | ±0.01 | ±0.01 | ±0.29 | ±0.09 | ±0.01 | ±0.00 | ±0.01 | ±0.01 | ±0.13 | ±0.06 | ±0.09 | ±0.03 | ±0.64 |

Table 2: OOD detection performance comparison of Medix and baselines on CIFAR-100 as InD data. Performance averaged over five runs; best results are highlighted in **bold**.

| Methods | SVHN | | PLACES365 | | LSUN-C | | LSUN-RESIZE | | TEXTURES | | Average | | InD ACC↑ |
|---|---|---|---|---|---|---|---|---|---|---|---|---|---|
| | FPR95↓ | AUROC↑ | FPR95↓ | AUROC↑ | FPR95↓ | AUROC↑ | FPR95↓ | AUROC↑ | FPR95↓ | AUROC↑ | FPR95↓ | AUROC↑ | |
| Using $\mathbb{P}_{in}$ only | | | | | | | | | | | | | |
| MSP | 84.59 | 71.44 | 82.84 | 73.78 | 66.54 | 83.79 | 82.42 | 75.38 | 83.29 | 73.34 | 79.94 | 75.55 | 75.96 |
| ODIN | 84.66 | 67.26 | 87.88 | 71.63 | 55.55 | 87.73 | 71.96 | 81.82 | 79.27 | 73.45 | 75.86 | 76.38 | 75.96 |
| Mahalanobis | 57.52 | 86.01 | 88.83 | 67.87 | 91.18 | 69.69 | 21.23 | 96.00 | 39.39 | 90.57 | 59.63 | 82.03 | 75.96 |
| Energy | 85.82 | 73.99 | 80.56 | 75.44 | 35.32 | 93.53 | 79.47 | 79.23 | 74.41 | 76.28 | 71.12 | 79.69 | 75.96 |
| KNN | 66.38 | 83.76 | 79.17 | 71.91 | 70.96 | 83.71 | 77.83 | 78.85 | 88.00 | 67.19 | 76.47 | 77.08 | 75.96 |
| ReAct | 74.33 | 88.04 | 81.33 | 74.32 | 39.30 | 91.19 | 79.86 | 73.69 | 67.38 | 82.80 | 68.44 | 82.01 | 75.96 |
| DICE | 88.35 | 72.58 | 81.61 | 75.07 | 26.77 | 94.74 | 80.21 | 78.50 | 76.29 | 76.07 | 70.65 | 79.39 | 75.96 |
| ASH | 21.36 | 94.28 | 68.37 | 71.22 | 15.27 | 95.65 | 68.18 | 85.42 | 40.87 | 92.29 | 42.81 | 87.77 | 75.96 |
| CSI | 64.70 | 84.97 | 82.25 | 73.63 | 38.10 | 92.52 | 91.55 | 63.42 | 74.70 | 92.66 | 70.26 | 81.44 | 69.90 |
| KNN+ | 32.21 | 93.74 | 68.30 | 75.31 | 40.37 | 88.98 | 44.86 | 88.88 | 46.26 | 87.40 | 46.40 | 86.29 | 73.78 |
| Using $\mathbb{P}_{in}$ and $\mathbb{P}_{wild}$ | | | | | | | | | | | | | |
| OE | 2.86 | 99.05 | 40.21 | 88.75 | 4.13 | 99.05 | 1.25 | 99.38 | 22.86 | 94.63 | 14.26 | 96.17 | 73.38 |
| Energy (w/ OE) | 2.71 | 99.34 | 34.82 | 90.05 | 3.27 | 99.18 | 2.54 | 99.23 | 30.15 | 94.76 | 14.70 | 96.51 | 72.76 |
| WOODS | 0.17 | 99.80 | 21.87 | 93.73 | 0.48 | 99.61 | 1.24 | 99.54 | 9.95 | 95.97 | 6.74 | 97.73 | 73.91 |
| Medix | **0.16** | **99.96** | **15.99** | **95.23** | **0.13** | **99.98** | **0.83** | **99.83** | **8.02** | **97.79** | **5.42** | **98.96** | 73.33 |
| (Ours) | ±0.02 | ±0.00 | ±0.66 | ±0.14 | ±0.06 | ±0.02 | ±0.36 | ±0.06 | ±0.75 | ±0.30 | ±0.37 | ±0.10 | ±0.83 |

WOODS by 7.21% on PLACES365 and 5.25% on TEXTURES in terms of FPR95, demonstrating its effectiveness in detecting OOD samples.

**A representative visual example of Medix.** We further investigate the performance of Algorithm 1 (Outlier Extraction) in extracting OOD samples from wild data $\mathcal{S}_{wild}$. To visualize this, we design an experiment using 2-dimensional synthetic data. This simulation is designed to be simple to facilitate better understanding. We generate the InD data by sampling from three multivariate Gaussian distributions, corresponding to three classes. The mean vectors are set to $[-2, 0]$, $[2, 0]$, and $[0, 2\sqrt{3}]$, respectively. The covariance matrix for all three classes is fixed at $0.25 \cdot I$. For the OOD data, we use a multivariate Gaussian distribution $\mathcal{N}([20, 2\sqrt{3}], 0.25 \cdot I)$. In Figure 2, we observe that our method successfully identifies outliers, the majority of which align with the ground truth. Medix successfully flags 87.5% of actual OOD samples as outliers, underscoring its robustness in outlier extraction.

**Additional studies and insights.** Due to space constraints, we defer additional experiments and insights to Appendix A, which include (1) ablation studies on hyperparameter selection and sensitivity analysis (Appendix A.2), showing Medix's strong robustness to hyperparameters, (2) a comparison between EWM and geometric median (Appendix A.1), showing that EWM is more sensitive to distributional shifts, making it more effective and reliable choice for filtering in our method, (3) a comprehensive comparison with methods employing competitive contrasting learning objectives (Appendix A.3), demonstrating that Medix outperforms even these competitive baselines by a significant margin, (4) evaluation on large-scale data under the complex unseen OOD setting, where $\mathbb{P}_{out}^{ptest} \neq \mathbb{P}_{out}$ (Appendix A.4), demonstrating that Medix outperforms baselines by a significant margin, (5) evaluating computation and memory efficiency of Medix (Appendix A.6) (6) evaluating the impact of pseudo-label quality (Appendix A.5), showing that our method is resilient to noisy or low-confidence labels, and (4) a comparison with semi-supervised open-set recognition methods (Appendix A.7).

## 6 RELATED WORK

In recent years, there has been a growing interest in OOD detection (Fort et al., 2021; Yang et al., 2024; Fang et al., 2022; Zhu et al., 2022; Yang et al., 2022; Wang et al., 2022c; Galil et al., 2023; Djurisic et al., 2023; Tao et al., 2023; Zheng et al., 2023; Wang et al., 2022b; 2023b; Uppal et al., 2023; Zhu et al., 2023; Bai et al., 2023; Ming & Li, 2024; Zhang et al., 2023; Ghosal et al., 2024). One

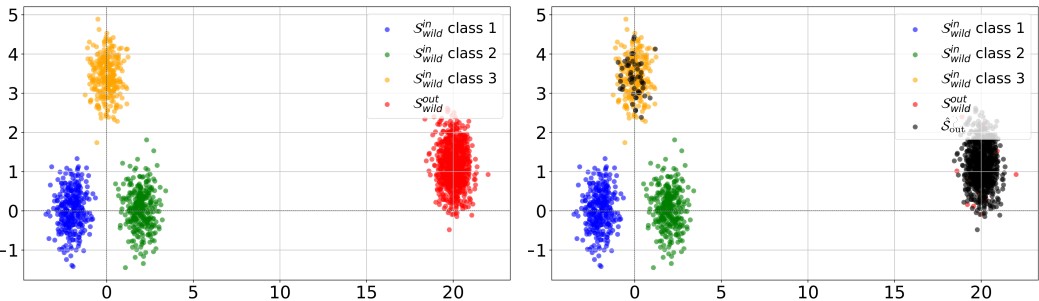

Figure 2: Example of Medix applied to unlabeled wild data. (a) Setup of the InD data $\mathcal{S}_{\text{wild}}^{\text{in}}$ and OOD data $\mathcal{S}_{\text{wild}}^{\text{out}}$ in the wild, with inliers sampled from three multivariate Gaussian distributions. (b) Outliers $\hat{\mathcal{S}}_{\text{out}}$ filtered by Medix (in black), with an error rate of $\hat{\mathcal{S}}_{\text{out}}$ containing InD data $\mathcal{S}_{\text{wild}}^{\text{in}}$ is only 12.5%.

approach to detect OOD data uses scoring functions to assess data distribution, including distance-based methods (Lee et al., 2018a; Tack et al., 2020a; Ren et al., 2021; Sehwag et al., 2021; Sun et al., 2022; Du et al., 2022a; Ming et al., 2023; Ren et al., 2022), gradient-based score (Huang et al., 2021b), energy-based score (Liu et al., 2020b; Wang et al., 2021; Wu et al., 2023), confidence-based approaches (Bendale & Boult, 2016; Hendrycks & Gimpel, 2017; Liang et al., 2018), and Bayesian methods (Gal & Ghahramani, 2016; Lakshminarayanan et al., 2017; Maddox et al., 2019; Malinin & Gales, 2019; Wen et al., 2020; Kristiadi et al., 2020).

Another approach to OOD detection involves using regularization techniques during the training phase (Malinin & Gales, 2018; Geifman & El-Yaniv, 2019; Hein et al., 2019; Meinke & Hein, 2020; Jeong & Kim, 2020; Liu et al., 2020a; Van Amersfoort et al., 2020; Yang et al., 2021; Wei et al., 2022; Du et al., 2022b; 2023; Wang et al., 2023a). For example, regularization techniques can be applied to the model to either reduce its confidence (Lee et al., 2017; Hendrycks et al., 2019) or increase its energy (Liu et al., 2020b; Du et al., 2022c; Ming et al., 2022) on the OOD data. Most of these regularization methods assume the availability of an auxiliary OOD dataset.

Several studies (Zhou et al., 2021; Katz-Samuels et al., 2022b; He et al., 2023) have relaxed the assumption of using only labeled data by incorporating unlabeled wild data (Katz-Samuels et al., 2022a; Geng et al., 2025), though they did not propose a clear mechanism for outlier detection. In contrast, Du et al. (2024a;b) introduced an explicit outlier filtering method, but their thresholding technique differs fundamentally from ours, as we utilize a new median-centric approach to detect the outliers. Additionally, Katz-Samuels et al. (2022a); Du et al. (2024a) operate under the assumption of batch-level mixing, where each batch has a set ratio of InD and OOD samples. However, with large outsourced datasets, such structured mixing is not available-data is mixed randomly across the dataset. Our method addresses this by enabling dataset-level mixing without relying on batch-level structure. Many studies also leverage positive-unlabeled learning, which trains classifiers using positive and/or unlabeled data (Letouzey et al., 2000; Hsieh et al., 2015; Plessis et al., 2015; Niu et al., 2016; Gong et al., 2018; Chapel et al., 2020; Garg et al., 2021; Xu & Denil, 2021; Garg et al., 2022; Zhao et al., 2022; Acharya et al., 2022). However, a key distinction from our approach is that these methods focus solely on differentiating $\mathbb{P}_{\text{out}}$ from $\mathbb{P}_{\text{in}}$, without simultaneously training an OOD classifier. Additionally, we propose a median-centric method to identify outliers in unlabeled data, with provably low error rates.

## 7 CONCLUSIONS

In this work, we introduced Medix, a novel median-centric framework for OOD detection that leverages unlabeled in-the-wild data. Using the inherent robustness of median operation, Medix effectively filters outliers from mixed unlabeled data, enabling the training of a reliable OOD detector. Our theoretical analysis established provable bounds on the inlier misclassification rate, demonstrating that Medix maintains robustness even under significant OOD contamination (up to 50%), with errors controlled by sub-Gaussian concentration and contamination effects. We also provided complementary theoretical limits on the rate of OOD misclassification to accurately isolate OOD samples under clear separation conditions. Empirical validation across diverse benchmarks showcased Medix's performance superiority over 20 baselines: it reduced the average FPR95 by 40.98% compared to strong baselines like KNN+ and outperformed state-of-the-art methods such as WOODS and DRL, while achieving an outlier extraction error rate as low as 12.5%.

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

# A ADDITIONAL STUDIES

## A.1 ELEMENT-WISE MEDIAN VS GEOMETRIC MEDIAN

Another pertinent question to ask is why we chose the element-wise median over the geometric median (Acharya et al., 2024) for outlier detection in Medix. To answer this question, we conducted a preliminary experiment using CIFAR-10 as the InD dataset and SVHN as the OOD dataset. Specifically, $\mathcal{S}_{\text{wild}}$ consists of 5k samples drawn from CIFAR-10, ensuring that these samples are disjoint from the training set used to train the model $\phi_{\mathcal{S}_{\text{in}}}$, which we leverage to compute $\bar{\nabla}_{\text{in}}$. We incrementally add SVHN OOD samples to $\mathcal{S}_{\text{wild}}$ and track the behavior of the $L_2$-norm deviation between $\bar{\nabla}_{\text{in}}$ and the element-wise median of the gradients of the wild dataset as well as the proportion of OOD samples removed. The results, shown in the Figure 3, demonstrate that for the same number of OOD samples in the wild dataset, the element-wise median identified a significantly higher proportion of OOD samples as outliers compared to the geometric median. This indicates that the element-wise median is more sensitive to distributional shifts, making it a more effective and reliable choice for filtering in our method.

Table 3: Effect of hyperparameters $\epsilon$ and $k$ on OOD detection.

| Method | FPR95↓ | $\epsilon$ | $k$ |
|---|---|---|---|
| DICE | 88.35 | – | – |
| ASH | 21.36 | – | – |
| CSI | 64.70 | – | – |
| KNN+ | 32.21 | – | – |
| OE | 2.86 | – | – |
| Energy | 2.71 | – | – |
| Medix | 0.16 | 0.005 | 20000 |
| Medix | 0.20 | 0.0005 | 20000 |
| Medix | 0.68 | 0.005 | 10000 |

## A.2 HYPERPARAMETER SELECTION AND SENSITIVITY ANALYSIS

We conducted an ablation study to assess the sensitivity of Medix to the hyperparameters $\epsilon$ and $k$. For this experiment, we employed CIFAR-100 as the InD dataset and SVHN as the OOD dataset. As shown in Table 3, Medix exhibits strong robustness to variations in the values of $\epsilon$ and $k$. In particular, Medix achieves a remarkably low FPR95 of 0.16 when using $\epsilon = 0.005$ and $k = 20000$. Even when the values of $\epsilon$ and $k$ are varied—e.g., reducing $\epsilon$ to 0.0005 or halving $k$ to 10000—the performance remains competitive (FPR95 of 0.20 and 0.68, respectively), demonstrating a graceful degradation rather than a sharp drop. The results indicate that while optimal hyperparameter selection is important, Medix maintains its effectiveness across a variety of hyperparameter choices, surpassing the baselines. This insensitivity to exact hyperparameter tuning makes Medix a reliable choice in real-world deployments, where exhaustive tuning may not be feasible.

## A.3 COMPREHENSIVE COMPARISON WITH RECENT COMPETITIVE METHODS

We have extended our comparisons to include competitive baselines that employ diverse contrasting learning objectives, such as SSD+ (Sehwag et al., 2021), ProxyAnchor (Kim et al., 2020), and CIDER Ming et al. (2023). Although these methods use contrasting learning objectives, which we do not, we included them for a comprehensive comparison. Additionally, we included more recent methods, such as CONJ (Peng et al., 2024), DRL (Zhang et al., 2024), Vim (Wang et al., 2022a), and VOS (Du et al., 2022c), to provide a more thorough evaluation. For this experiment, we use CIFAR-100 as InD

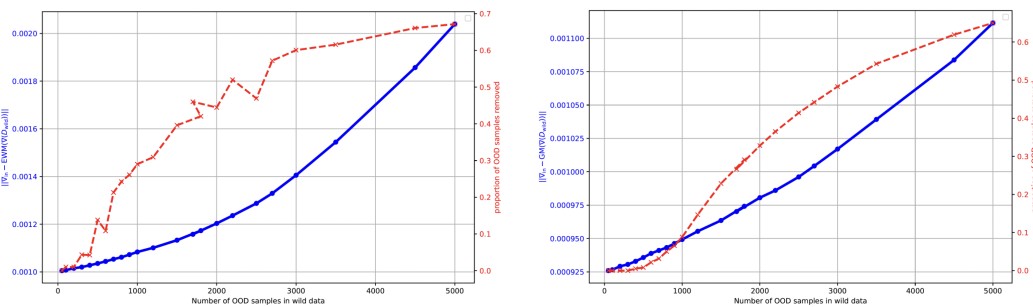

Figure 3: Comparison of element-wise median (EWM) and geometric median (GM).

Table 4: OOD detection performance comparison of Medix and recent competitive baselines on CIFAR-100 as InD data. Performance averaged over five runs; best results are highlighted in **bold**.

| Methods | OOD Datasets | | | | | | | | | |
| --- | --- | --- | --- | --- | --- | --- | --- | --- | --- | --- |
| | SVHN | | PLACES365 | | LSUN-C | | TEXTURES | | AVERAGE | |
| | FPR95↓ | AUROC↑ | FPR95↓ | AUROC↑ | FPR95↓ | AUROC↑ | FPR95↓ | AUROC↑ | FPR95↓ | AUROC↑ |
| MSP | 78.89 | 79.80 | 84.38 | 74.21 | 83.47 | 75.28 | 86.51 | 72.53 | 83.31 | 75.46 |
| Mahalanobis | 87.09 | 80.62 | 84.63 | 73.89 | 84.15 | 79.43 | 61.72 | 84.87 | 79.40 | 79.70 |
| ODIN | 70.16 | 84.88 | 82.16 | 75.19 | 76.36 | 80.10 | 85.28 | 75.23 | 78.49 | 78.85 |
| Energy | 66.91 | 85.25 | 81.41 | 76.37 | 59.77 | 86.69 | 79.01 | 79.96 | 71.78 | 82.07 |
| ReAct | 50.93 | 88.75 | 83.55 | 73.10 | 64.02 | 80.31 | 64.40 | 81.95 | 65.72 | 81.03 |
| KNN | 46.25 | 90.39 | 82.08 | 75.44 | 60.85 | 85.61 | 62.39 | 83.95 | 62.89 | 83.85 |
| Vim | 73.42 | 84.62 | 85.34 | 69.34 | 86.96 | 69.74 | 74.56 | 76.23 | 80.07 | 74.98 |
| VOS | 43.24 | 82.80 | 76.85 | 78.63 | 73.61 | 84.69 | 57.57 | 87.31 | 62.82 | 83.36 |
| CSI | 44.53 | 92.65 | 79.08 | 76.27 | 75.58 | 83.78 | 61.61 | 86.47 | 65.20 | 84.79 |
| ProxyAnchor | 87.21 | 82.43 | 70.10 | 79.84 | 37.19 | 91.68 | 65.64 | 84.99 | 65.04 | 84.74 |
| SSD+ | 31.19 | 94.19 | 77.74 | 79.90 | 79.39 | 85.18 | 66.63 | 86.18 | 63.74 | 86.36 |
| KNN+ | 39.23 | 92.78 | 80.74 | 77.58 | 48.99 | 89.30 | 57.15 | 88.35 | 56.53 | 87.00 |
| ASH | 52.96 | 90.19 | 72.62 | 76.38 | 75.18 | 76.52 | 56.17 | 86.75 | 64.23 | 82.46 |
| CIDER | 23.09 | 95.16 | 79.63 | 73.43 | 16.16 | 96.33 | 43.87 | 90.42 | 40.69 | 88.84 |
| CONJ | 46.19 | 90.44 | 80.81 | 75.83 | 60.45 | 85.90 | 62.13 | 83.77 | 62.40 | 83.99 |
| DRL | 20.15 | 94.07 | 76.64 | 77.55 | 16.97 | 94.63 | 31.97 | 92.09 | 36.43 | 89.59 |
| Medix (Ours) | **0.48** ±0.05 | **99.87** ±0.02 | **24.52** ±1.76 | **93.42** ±0.36 | **0.73** ±0.08 | **99.84** ±0.03 | **8.99** ±0.72 | **97.92** ±0.24 | **8.68** ±0.65 | **97.76** ±0.16 |

dataset and train the ResNet-34 model[1] following the setup in Ming et al. (2023); Zhang et al. (2024). The model is trained using stochastic gradient descent with momentum 0.9, and weight decay $10^{-4}$ for 500 epochs. The initial learning rate is 0.5 with cosine scheduling and the batch size is 512. We use the checkpoints provided by Ming et al. (2023)[2]. The results, as shown in Table 4, demonstrate that Medix outperforms even these competitive baselines by a significant margin, including those trained with contrastive learning objectives, achieving an average FPR95 of 8.68% and an average AUROC of 97.76%. Furthermore, Medix provides a provably low error rate, offering theoretical guarantees that most alternative approaches do not.

## A.4 EVALUATION ON LARGE-SCALE, COMPLEX UNSEEN OOD SETTINGS

We next investigate whether Medix can handle large-scale wild OOD data under the unseen OOD setting. This setting is more challenging for two main reasons: (1) it leverages a large-scale wild OOD dataset ($\mathbb{P}_{\text{out}}$), increasing both data complexity and computational demand, and (2) the test-time OOD distribution ($\mathbb{P}_{\text{out}}^{\text{test}}$) is intentionally chosen to be distributionally different from the wild OOD data used during training, i.e., $\mathbb{P}_{\text{out}}^{\text{test}} \neq \mathbb{P}_{\text{out}}$, which better reflects realistic and challenging deployment scenarios.

We used CIFAR-100 as the labeled InD data, 300K Random Images dataset Hendrycks et al. (2019) as the unlabeled wild OOD data and tested on the SVHN dataset as the test OOD data. This setup introduces a greater level of complexity because the large-scale wild OOD data (300K Random Images) is significantly different from the OOD test data (SVHN). To evaluate the performance of our approach, we compare it against baselines that also leverage wild data in training. The results in the Table 5 highlight the superior performance of our method, Medix, compared to the baselines, with a significantly lower FPR95 ($41.29 \pm 1.2$) and higher AUROC ($87.25 \pm 0.6$), showing its effectiveness in distinguishing between InD and OOD data.

Table 5: Comparison of OOD detection performance on large-scale unseen OOD data.

| Method | FPR ↓ | AUROC ↑ |
| --- | --- | --- |
| OE | $68.80 \pm 2.8$ | $82.89 \pm 1.1$ |
| Energy (w/ OE) | $69.81 \pm 2.4$ | $85.59 \pm 1.0$ |
| WOODS | $69.41 \pm 2.7$ | $86.76 \pm 0.8$ |
| Medix (ours) | $\mathbf{41.29 \pm 1.2}$ | $\mathbf{87.25 \pm 0.9}$ |

---

[1]Since ResNet-34 is the standard model across these works, we adopt the same model to ensure consistency and facilitate a fair comparison.

[2]https://github.com/deeplearning-wisc/cider

## A.5 EVALUATING THE IMPACT OF PSEUDO-LABEL QUALITY ON OOD FILTERING AND ROBUSTNESS

Another important question to ask is whether the quality of pseudo-labels $\hat{y}_{\tilde{x}_i}$, particularly in terms of softmax confidence, affects the performance of OOD filtering, and whether a simple pre-filtering step that removes low-confidence pseudo-labels could improve the robustness of the model. To answer this question, we conducted an experiment to evaluate how filtering low-confidence pseudo-labels affects OOD filtering. For this experiment, we used CIFAR-10 as the InD dataset and LSUN-Resize as the OOD dataset. Specifically, we computed the softmax probabilities for each pseudo-labeled sample and discarded those with low-confidence predictions, setting a threshold of 0.6. After filtering, we found that 15.98% of the samples were removed from the training set.

The results showed that there was virtually no difference in performance between the method with filtering and the method without filtering. Both methods yielded a FPR95 of 0.01%, but with filtering, AUROC increased slightly from 99.98% to 99.99%. Thus, removing low-confidence pseudo-labels doesn't significantly impact the robustness of the model or its ability to detect OOD samples, showing that our method is resilient to noisy or low-confidence labels, and further filtering steps are unlikely to yield meaningful improvements.

## A.6 EVALUATING COMPUTATION AND MEMORY EFFICIENCY IN MEDIX FILTERING

We now turn to the question of whether Medix's filtering phase is computationally feasible and memory-efficient on large datasets. In response to this question, we conducted profiling experiments on an NVIDIA A100-SXM4-80GB GPU using approximately 15,000 unlabeled samples for each InD–OOD pair. For this experiment, we used CIFAR-10 and CIFAR-100 as the InD datasets and LSUN-Resize as the OOD dataset. As seen from the results in Table 6, the GPU memory usage remains modest, and the filtering phase completes within a tractable timeframe, even when handling thousands of unlabeled samples. This demonstrates that the Medix filtering process is computationally feasible and does not impose excessive memory overhead, even on large datasets.

Table 6: Profiling results for Medix filtering on NVIDIA A100 80GB GPU.

| InD–OOD | Wall Clock Time (s) | Peak GPU Memory (MB) | Current GPU Memory (MB) |
|---|---|---|---|
| CIFAR10 – LSUN-Resize | 4497.17 | 99.46 | 31.74 |
| CIFAR100 – LSUN-Resize | 5478.36 | 99.56 | 31.79 |

## A.7 COMPARISON WITH SEMI-SUPERVISED OPEN-SET RECOGNITION METHODS

Our work differs from recent semi-supervised open-set recognition methods (Saito et al., 2021; Fan et al., 2023; Hang & Zhang, 2024; Wang et al., 2024) in several key ways. For example, in OpenML Saito et al. (2021), the main problem is to handle outliers in semi-supervised learning (SSL) when training a standard classifier, whereas in our setting, the main challenge is to detect OOD samples from unlabeled wild data and train a dedicated OOD detector classifier. In other words, while SSL methods (Saito et al., 2021; Fan et al., 2023; Hang & Zhang, 2024; Wang et al., 2024) aim to improve classification despite outliers, our approach enhances OOD detection in an open-world setting where labeled OOD data is unavailable.

Secondly, these SSL methods aim to train a classifier that is robust to OOD samples in SSL, treating outliers as noise, while in our setting, the goal is to explicitly detect OOD samples and train an OOD detector. In other words, SSL methods focus on suppressing OOD samples during training to improve SSL performance, whereas Medix actively detects and leverages these OOD samples to build a more effective and reliable OOD detection system.

Lastly, the two approaches differ in their techniques: OpenMatch is based on consistency regularization, where the main idea is to enforce consistency across different stochastic transformations of the same input. This is mathematically modeled through soft constraints on the classifier's outputs, encouraging smooth decision boundaries. On the other hand, Medix relies on median-based filtering for outlier detection. The method uses statistical robustness properties of the median, leveraging its insensitivity to extreme values to identify potential OOD samples. The theoretical analysis derives

error bounds based on the contamination effect (proportion of OOD samples) and the concentration effect (sub-Gaussian behavior of InD gradients), ensuring that OOD detection error remains low.

## B  BROADER IMPACTS AND LIMITATIONS

As machine learning continues to advance, tackling the challenges of OOD detection has become essential for ensuring robust and reliable model performance in real-world applications. We introduce a median-centric framework for OOD detection that enhances OOD handling and model safety. The impact of our research goes beyond theoretical advancements, with practical applications in healthcare, autonomous systems, and finance. By improving OOD detection, we address a key challenge in model deployment, fostering greater trust and adoption of machine learning technologies. Future work could explore integrating it with generative models for outlier synthesis or adapting it to dynamic environments where OOD distributions evolve over time.

A potential limitation of Medix lies in its reliance on a mixture of unlabeled InD and OOD data, which, in practice, may be noisy, corrupted, or inconsistently sampled. However, this assumption is not unique to our method—it reflects the inherent uncertainty of real-world deployment environments and is a common starting point in robust learning theory. To make our assumptions explicit and verifiable, we rigorously formalize them in the main theorems and provide a precise geometric condition in Remark 4.3 to justify the necessary separation required for reliable filtering. Moreover, we extend our guarantees to looser settings in Theorem C.3, removing the need for sub-Gaussian tails and demonstrating the resilience of our framework under weaker distributional assumptions. These efforts underscore that while Medix does make simplifying assumptions, they are both interpretable and relaxable, forming a principled foundation for future improvements in OOD detection under realistic noise conditions.

## C   PROOF OF THEOREM

To aid clarity and consistency throughout the theoretical and empirical sections, we summarize the key notation used in this paper in Table 7. This includes definitions for gradient-related quantities, sample sizes, and filtering terms, as well as parameters that appear in our concentration bounds. Unless stated otherwise, we use boldface for vectors, $\mathcal{X}$ to denote the input space, and assume all gradients are computed with respect to a fixed pretrained model $f_\phi$.

Table 7: Notation summary. We use bold symbols for vectors and $\mathcal{X}$ to denote the input space.

| Symbol | Meaning |
|---|---|
| $\mathcal{X}$ | Input space (e.g., images in $[0,1]^d$) |
| $f_\phi$ | Model parameterized by $\phi$ |
| $\ell(f_\phi(x))$ | Loss function evaluated at input $x$ |
| $\nabla\ell(f_\phi(x))$ | Gradient of loss at point $x$ |
| $\mathbf{g}(x)$ | Shorthand for $\nabla\ell(f_\phi(x))$, gradient vector |
| $\bar{\mathbf{g}}_{\text{in}}$ | Empirical mean of gradients over InD points |
| $m$ | Total number of unlabeled samples in $\mathcal{S}_{\text{wild}}$ |
| $m_{\text{in}} = (1-\pi)m$ | Number of InD samples |
| $m_{\text{out}} = \pi m$ | Number of OOD samples |
| $\pi \in (0,1)$ | OOD contamination ratio |
| $\mathcal{S}_{\text{wild}}$ | Unlabeled mixture of InD and OOD data |
| $\mathcal{S}_{\text{in}}^\star$ | Inliers selected by the median filter |
| $\mathcal{S}_{\text{out}}^\star$ | OOD points selected by the median filter |
| $\mathcal{B}_\varepsilon(\bar{\mathbf{g}}_{\text{in}})$ | Ball of radius $\varepsilon$ centered at $\bar{\mathbf{g}}_{\text{in}}$ |
| $m_\varepsilon$ | Number of InD points inside $\mathcal{B}_\varepsilon$ |
| $m_\varepsilon^\star$ | Number of selected inliers inside $\mathcal{B}_\varepsilon$ |
| $m_{\text{sep}}^\star$ | Number of well-separated OODs in selected set |
| $\text{ERR}_{\text{in}}$ | Fraction of InD points misclassified as OOD |
| $\text{ERR}_{\text{out}}$ | Fraction of OOD points misclassified as InD |
| $\sigma^2$ | Sub-Gaussian proxy variance of each gradient coordinate |
| $\varepsilon$ | Threshold for gradient deviation, $\varepsilon = \sigma\sqrt{2\log(2dm_{\text{in}})}$ |
| $\delta \in (0,1)$ | Confidence parameter for concentration bounds |
| $d$ | Dimensionality of gradient vectors |

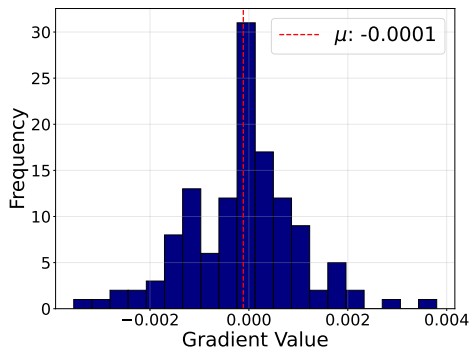

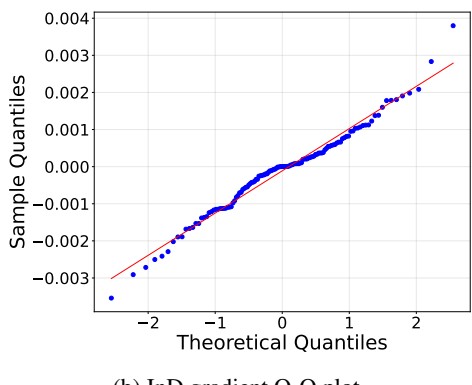

(a) InD gradient histogram.            (b) InD gradient Q-Q plot.

Figure 4: Illustration of InD sample gradients exhibiting sub-Gaussian behavior in each coordinate. (left) Histogram of gradient values (CIFAR-100 InD data) showing concentration around the mean with light tails, consistent with sub-Gaussianity. (right) Q-Q plot comparing empirical quantiles of InD gradients against a theoretical Gaussian distribution, confirming alignment with sub-Gaussianity.

## C.1 Bound on the Inlier Misclassification Rate

**Theorem C.1** (Inlier Misclassification Bound). *Assume that the gradients of InD points in $\mathcal{S}_{\text{wild}}$ are i.i.d., and each coordinate is sub-Gaussian with variance proxy $\sigma^2$. Let $\epsilon = \sigma\sqrt{2\log(2dm_{\text{in}})}$, and fix any confidence level $\delta \in (0, 1)$. Then, with probability at least $1 - \delta$, the inlier misclassification rate of the element-wise median (EWM) filtering rule satisfies:*

$$\text{ERR}_{\text{in}} \leq \underbrace{\frac{1}{m_{in}} + 2\sqrt{\frac{\log(1/\delta)}{2m_{\text{in}}}}}_{\text{Concentration term}} + \underbrace{\frac{\pi}{2(1-\pi)}}_{\text{Contamination term}} .$$

*Proof.* We structure our argument into two key steps. In **Step 1**, we establish a high-probability concentration bound for the gradients of InD samples around their mean, denoted as $\bar{\nabla}_{\text{in}}$. We define an InD sample whose gradient lies within this high-probability region as a *good inlier*. In **Step 2**, we leverage a "swapping" argument on the EWM objective to demonstrate that removing more than a $(2\eta + \frac{\pi}{2}(1-\pi))$ fraction of *good inliers*—where $\eta$ represents the fraction of InD samples with gradients outside the high-probability region (i.e., non-good InD samples)—would contradict optimality. Combining these two steps, we derive the stated upper bound on the inlier misclassification rate, $\text{ERR}_{\text{in}}$.

**Step 1: Concentration of InD Gradients.** Let $\nabla\ell(f_\phi(\boldsymbol{x})) \in \mathbb{R}^d$ be the gradient (with respect to $\phi$) of a loss evaluated at an InD sample $\boldsymbol{x}$. Suppose each coordinate of this gradient is sub-Gaussian with variance proxy $\sigma^2$, centered at the same mean vector $\bar{\nabla}_{\text{in}}$. That is, for each coordinate $j \in \{1, \ldots, d\}$,

$$\mathbb{P}\Big(\big|\nabla\ell(f_\phi(\boldsymbol{x}))_j - \bar{\nabla}_{\text{in},j}\big| \geq \epsilon\Big) \leq 2\exp\Big(-\tfrac{\epsilon^2}{2\sigma^2}\Big). \tag{6}$$

Define

$$v := \nabla\ell(f_\phi(\boldsymbol{x})) - \bar{\nabla}_{\text{in}} \in \mathbb{R}^d.$$

It is straightforward to show that

$$\|v\|_2 \leq \sqrt{d}\,\|v\|_\infty,$$

so if $\|v\|_2 > \epsilon\sqrt{d}$ then there must exists at least one coordinate $j$ with $|v_j| > \epsilon$.

Hence, combining equation 6 across $j = 1, \ldots, d$ via a union bound yields

$$\mathbb{P}\Big(\|\nabla\ell(f_\phi(\boldsymbol{x})) - \bar{\nabla}_{\text{in}}\|_2 > \epsilon\sqrt{d}\Big) \leq 2d\exp\Big(-\tfrac{\epsilon^2}{2\sigma^2}\Big). \tag{7}$$

Let $m_{\text{in}} = (1-\pi)m$ denote the total number of InD points in the set $\mathcal{S}_{\text{wild}}$. We define an InD point $\boldsymbol{x}$ to be *bad* if

$$\|\nabla\ell(f_\phi(\boldsymbol{x})) - \bar{\nabla}_{\text{in}}\|_2 > \epsilon\sqrt{d}. \tag{8}$$

Our goal is to show that, with high probability, only a small fraction of these $m_{\text{in}}$ inliers can be bad.

From our sub-Gaussian assumption and applying the union bound across the $d$ coordinates, we define a constant $p \in [0, 1]$ as follows:

$$p := \mathbb{P}\left(\|\nabla\ell(f_\phi(\boldsymbol{x})) - \bar{\nabla}_{\text{in}}\|_2 > \epsilon\sqrt{d}\right) \leq 2d\exp\left(-\frac{\epsilon^2}{2\sigma^2}\right). \tag{9}$$

Here, $\boldsymbol{x}$ is a generic InD sample, and we assume that the samples are i.i.d. across the $m_{\text{in}}$ inliers.

Let $X$ be the random variable denoting the number of bad inliers among the $m_{\text{in}}$ samples. For each inlier $\boldsymbol{x}_i$, define the indicator variable

$$I_i := \begin{cases} 1, & \text{if } \|\nabla\ell(f_\phi(\boldsymbol{x})) - \bar{\nabla}_{\text{in}}\|_2 > \epsilon\sqrt{d}, \\ 0, & \text{otherwise.} \end{cases} \tag{10}$$

Then, $X$ can be expressed as

$$X = \sum_{i=1}^{m_{\text{in}}} I_i. \tag{11}$$

By the linearity of expectation, we have

$$\mathbb{E}[X] = \sum_{i=1}^{m_{\text{in}}} \mathbb{E}[I_i] \tag{12}$$

$$= \sum_{i=1}^{m_{\text{in}}} \mathbb{P}(\boldsymbol{x}_i \text{ is bad}) \tag{13}$$

$$\stackrel{\text{from equation 9}}{\leq} m_{\text{in}} \cdot p \tag{14}$$

$$\leq m_{\text{in}} \cdot 2d \exp\left(-\frac{\epsilon^2}{2\sigma^2}\right). \tag{15}$$

Observing that $X$ is a sum of $m_{\text{in}}$ Bernoulli($p$) random variables, each bounded between 0 and 1, we can apply Hoeffding's inequality to obtain, for any $t > 0$,

$$\mathbb{P}\left(X \geq \mathbb{E}[X] + t\right) \leq \exp\left(-\frac{2t^2}{m_{\text{in}}}\right). \tag{16}$$

Setting

$$t := \sqrt{\frac{m_{\text{in}} \log\left(\frac{1}{\delta}\right)}{2}}, \tag{17}$$

we obtain

$$\mathbb{P}\left(X \geq \mathbb{E}[X] + \sqrt{\frac{m_{\text{in}} \log(1/\delta)}{2}}\right) \leq \delta. \tag{18}$$

With probability at least $1 - \delta$, we therefore have

$$X \leq \mathbb{E}[X] + \sqrt{\frac{m_{\text{in}} \log(1/\delta)}{2}}. \tag{19}$$

Dividing both sides by $m_{\text{in}}$ yields an upper bound on the fraction of bad inliers:

$$\frac{X}{m_{\text{in}}} \leq \frac{\mathbb{E}[X]}{m_{\text{in}}} + \sqrt{\frac{\log(1/\delta)}{2m_{\text{in}}}} \tag{20}$$

$$\stackrel{\text{from equation 12}}{\leq} 2d \exp\left(-\frac{\epsilon^2}{2\sigma^2}\right) + \sqrt{\frac{\log(1/\delta)}{2m_{\text{in}}}} \tag{21}$$

$$= \underbrace{\frac{1}{m_{\text{in}}} + 2\sqrt{\frac{\log(1/\delta)}{2m_{\text{in}}}}}_{\text{Concentration term}} + \underbrace{\frac{\pi}{2(1-\pi)}}_{\text{Contamination term}}. \tag{22}$$

Let us denote this fraction by $\eta$. By setting $\epsilon = \sigma\sqrt{2\log(2dm_{\text{in}})}$ Thus, *with probability at least $1 - \delta$, no more than an $O\left(d \exp\left(-\frac{\epsilon^2}{2\sigma^2}\right) + \sqrt{\frac{\log(1/\delta)}{m_{\text{in}}}}\right)$ fraction of inliers have gradients lying outside the radius $\epsilon\sqrt{d}$ around $\bar{\nabla}_{\text{in}}$.* These are precisely the *bad inliers*, while the remainder are called *good inliers*.

**Step 2: Contradiction via a Swapping Argument** We demonstrate that if $\mathcal{S}^*$ excludes an excessive number of InD, specifically when

$$\mathrm{ERR}_{\mathrm{in}} > 2\eta + \frac{\pi}{2(1-\pi)}, \tag{23}$$

then it is possible to construct a new set $\widehat{\mathcal{S}}$ whose EWM is strictly closer to $\bar{\nabla}_{\mathrm{in}}$ than that of $\mathcal{S}^*$, thereby contradicting the optimality of $\mathcal{S}^*$.

Let $m_\varepsilon$ be the number of indices $i$ such that either $x_i$ is OOD or $x_i$ is an InD point whose gradient lies outside the closed $\ell_2$ ball

$$B_\varepsilon = \left\{ g \in \mathbb{R}^d : \|g - \bar{\nabla}_{\mathrm{in}}\|_2 \leq \varepsilon\sqrt{d} \right\}. \tag{24}$$

Let $m_{\mathrm{in}}^*$ be the number of InD points contained in the candidate minimizer $\mathcal{S}^*$.

Assume for contradiction that

$$\mathrm{ERR}_{\mathrm{in}} > 2\eta + \frac{\pi}{2(1-\pi)}, \tag{25}$$

Define the swap size as

$$m_\varepsilon' := \min\left\{ m_\varepsilon, (\mathrm{ERR}_{\mathrm{in}} - \eta)m_{\mathrm{in}} \right\}. \tag{26}$$

This quantity counts how many good inliers (those inside $B_\varepsilon$) are currently excluded from $\mathcal{S}^*$ and available for swapping in.

Construct a new set $\widehat{\mathcal{S}}$ by removing $m_\varepsilon'$ points from $\mathcal{S}^*$ that lie outside $B_\varepsilon$ (OOD points or bad inliers) and inserting $m_\varepsilon'$ good inliers from $B_\varepsilon$ in their place. This maintains cardinality:

$$|\widehat{\mathcal{S}}| = |\mathcal{S}^*|.$$

Recall that

$$\mathrm{ERR}_{\mathrm{in}} := \frac{m_\varepsilon - m_{\mathrm{in}}^*}{m_{\mathrm{in}}}, \tag{27}$$

which measures the fraction of good inliers excluded from $\mathcal{S}^*$. Rearranging, we obtain:

$$m_\varepsilon - m_{\mathrm{in}}^* = \mathrm{ERR}_{\mathrm{in}} \cdot m_{\mathrm{in}}. \tag{28}$$

Now we know, for contradiction, that

$$\mathrm{ERR}_{\mathrm{in}} > 2\eta + \frac{\pi}{2(1-\pi)}, \tag{29}$$

Subtracting $\eta$ from both sides and multiplying by $m_{\mathrm{in}}$, we obtain:

$$(\mathrm{ERR}_{\mathrm{in}} - \eta)m_{\mathrm{in}} > \left( \eta + \frac{\pi}{2(1-\pi)} \right) m_{\mathrm{in}}. \tag{30}$$

We know that at most a $\eta$ fraction of inliers are outside $B_\varepsilon$, and the fraction of OOD points among all samples is $\pi$. Therefore, the total number of samples in the dataset that lie outside $B_\varepsilon$ satisfies:

$$m_\varepsilon \leq \eta m_{\mathrm{in}} + \frac{\pi}{1-\pi} m_{\mathrm{in}} = \left( \eta + \frac{\pi}{1-\pi} \right) m_{\mathrm{in}}. \tag{31}$$

It follows that

$$\left( \eta + \frac{\pi}{2(1-\pi)} \right) m_{\mathrm{in}} > \frac{1}{2} \cdot \left( \eta + \frac{\pi}{1-\pi} \right) m_{\mathrm{in}} \geq \frac{m_\varepsilon}{2}. \tag{32}$$

Combining with the previous inequality gives:

$$(\text{ERR}_{\text{in}} - \eta)m_{\text{in}} > \frac{m_\varepsilon}{2}, \tag{33}$$

So $m'_\varepsilon = m_\varepsilon/2 + \eta m_{\text{in}}$, which implies

$$m_\varepsilon - m'_\varepsilon = \frac{m_\varepsilon}{2} - \eta m_{\text{in}}, \tag{34}$$

$$m^*_{\text{in}} + m'_\varepsilon = m^*_{\text{in}} + \frac{m_\varepsilon}{2} + \eta m_{\text{in}}. \tag{35}$$

Because $\eta m_{\text{in}} > 0$, it follows that

$$m^*_{\text{in}} + m'_\varepsilon > m_\varepsilon - m'_\varepsilon, \tag{36}$$

so the points lying inside $B_\varepsilon$ form a strict majority in $\widehat{\mathcal{S}}$.

Fix any coordinate $j \in \{1, \ldots, d\}$. Let $X^{(j)} = \{x_1^{(j)}, \ldots, x_m^{(j)}\}$ be the $j$-th coordinate values of all gradients in $\mathcal{S}^*$ and $\widehat{X}^{(j)} = \{\hat{x}_1^{(j)}, \ldots, \hat{x}_m^{(j)}\}$ the same in $\widehat{\mathcal{S}}$. Let $\mu_j = \bar{\nabla}_{\text{in},j}$.

Each inserted gradient lies inside the interval $I_j = [\mu_j - \varepsilon, \mu_j + \varepsilon]$, and each removed gradient lies outside $I_j$. So for some coordinate $j^*$, the number of values inside $I_{j^*}$ increases from below $m/2$ to above $m/2$ due to the swaps. Therefore:

$$\text{med}(X^{(j^*)}) \notin I_{j^*}, \qquad \text{med}(\widehat{X}^{(j^*)}) \in I_{j^*}, \tag{37}$$

which implies

$$|\text{med}(\widehat{X}^{(j^*)}) - \mu_{j^*}| \le \varepsilon \quad \text{and} \quad |\text{med}(\widehat{X}^{(j^*)}) - \mu_{j^*}| < |\text{med}(X^{(j^*)}) - \mu_{j^*}|. \tag{38}$$

For the remaining coordinates $j \neq j^*$, the swapped points either leave the majority unchanged or increase it, so the median distance from $\mu_j$ does not increase and possibly decreases.

Thus, component-wise:

$$|\text{EWM}_j(G_{\widehat{\mathcal{S}}}) - \mu_j| \le |\text{EWM}_j(G_{\mathcal{S}^*}) - \mu_j|, \tag{39}$$

with strict inequality in at least one coordinate $j^*$, implying

$$\|\text{EWM}(G_{\widehat{\mathcal{S}}}) - \bar{\nabla}_{\text{in}}\|_2 < \|\text{EWM}(G_{\mathcal{S}^*}) - \bar{\nabla}_{\text{in}}\|_2. \tag{40}$$

This contradicts the assumption that $\mathcal{S}^*$ is a minimizer of the optimization problem, which concludes the proof.

$$\Rightarrow \quad \text{ERR}_{\text{in}} \le 2\eta + \frac{\pi}{2(1-\pi)}. \tag{41}$$

$\square$

## C.2 BOUND ON THE OUTLIER RETENTION RATE

We now establish a non-asymptotic upper bound on the fraction of OOD points that remain in the subset returned by the MEDIX filter. This result mirrors the inlier misclassification bound (Appendix C.1), but instead of relying on inlier concentration, it exploits a separation assumption between the OOD and InD gradient means.

**Theorem C.2** (Outlier Retention Bound under Vector Separation). *Assume that for every OOD point $x \sim P_{\text{out}}$, the gradient $g(x) := \nabla\ell(f_{\varphi_{\text{in}}}(x)) \in \mathbb{R}^d$ satisfies:*

*(I) Sub-Gaussianity. Each coordinate $g_j(x)$ is sub-Gaussian with variance proxy $\sigma_{\text{out}}^2$.*

*(II) Vector Mean Separation. The mean OOD gradient vector $\mu_{\text{out}}$ satisfies*

$$\left\|\mu_{\text{out}} - \bar{\nabla}_{\text{in}}\right\|_2 \geq \Delta\sqrt{d}, \quad \text{for some } \Delta > 0.$$

*Fix any tolerance $\varepsilon \in (0, \Delta)$ and confidence parameter $\delta \in (0, 1)$. Define:*

$$p_{\text{out}}(\varepsilon) := 2d\exp\left(-\frac{(\Delta - \varepsilon)^2}{2\sigma_{\text{out}}^2}\right), \qquad \eta_{\text{out}} := p_{\text{out}}(\varepsilon) + \sqrt{\frac{\log(1/\delta)}{2m_{\text{out}}}}.$$

*Then with probability at least $1 - \delta$, the outlier retention rate satisfies:*

$$\text{ERR}_{\text{out}} := \frac{|\mathcal{S}_{\text{in}}^\star \cap \text{OOD}|}{m_{\text{out}}} \leq 2d\exp\left(-\frac{(\Delta - \varepsilon)^2}{2\sigma_{\text{out}}^2}\right) + \sqrt{\frac{\log(1/\delta)}{2m_{\text{out}}}} + \frac{1 - \pi}{2\pi}. \tag{42}$$

*Proof.* **Step 1: Concentration of OOD Gradients.**
By assumption, each coordinate of $g(x)$ is sub-Gaussian, and the mean vector is separated from $\bar{\nabla}_{\text{in}}$ by at least $\Delta\sqrt{d}$. Therefore, for any OOD point $x$, the event

$$\left\|g(x) - \bar{\nabla}_{\text{in}}\right\|_2 \leq \varepsilon\sqrt{d} \tag{43}$$

has probability at most $p_{\text{out}}(\varepsilon)$.

**Counting the "ambiguous" OOD points.** For every OOD example $x_i$ define the indicator

$$J_i := \mathbf{1}\left\{\|g(x_i) - \bar{\nabla}_{\text{in}}\|_2 \leq \varepsilon\sqrt{d}\right\}, \qquad i = 1, \ldots, m_{\text{out}}.$$

Each $J_i \sim \text{Bernoulli}(p_{\text{out}}(\varepsilon))$ and the $J_i$'s are independent. Let

$$Y := \sum_{i=1}^{m_{\text{out}}} J_i, \qquad \mu := \mathbb{E}[Y] = m_{\text{out}}\, p_{\text{out}}(\varepsilon).$$

Because the indicators $J_i$ take values in $\{0, 1\}$, Hoeffding's inequality states that for any $s > 0$

$$\Pr\{Y - \mu \geq s\} \leq \exp\left(-\frac{2s^2}{m_{\text{out}}}\right).$$

Choose $s = \sqrt{\frac{m_{\text{out}}\log(1/\delta)}{2}}$; then equation C.2 equals $\delta$, and with probability at least $1 - \delta$

$$Y \leq m_{\text{out}}\left(p_{\text{out}}(\varepsilon) + \sqrt{\frac{\log(1/\delta)}{2m_{\text{out}}}}\right) = m_{\text{out}}\eta_{\text{out}}.$$

Define the high-probability event

$$\mathcal{E}_1 := \{Y \leq m_{\text{out}}\eta_{\text{out}}\}, \tag{35}$$

which holds with probability at least $1 - \delta$.

**Step 2: Contradiction via a Swapping Argument** Suppose, toward contradiction, that

$$\text{ERR}_{\text{out}} > \eta_{\text{out}} + \frac{1 - \pi}{2\pi}. \tag{44}$$

Let $m_{\text{out}}^\star := |\mathcal{S}_{\text{in}}^\star \cap \text{OOD}|$, and define $m_\varepsilon^\star := \#\{\text{ambiguous OODs in } \mathcal{S}_{\text{in}}^\star\}$. Then, under $\mathcal{E}_1$,

$$m_{\text{sep}}^\star := m_{\text{out}}^\star - m_\varepsilon^\star > \frac{1-\pi}{2\pi}m_{\text{out}} = \frac{1}{2}m_{\text{in}}. \tag{45}$$

Thus, more than half the points in $\mathcal{S}_{\text{in}}^\star$ are separated OODs whose gradients lie outside the ball

$$B_\varepsilon := \left\{g \in \mathbb{R}^d : \left\|g - \bar{\nabla}_{\text{in}}\right\|_2 \leq \varepsilon\sqrt{d}\right\}. \tag{46}$$

By the definition of the coordinate-wise median, this implies that in every coordinate, the majority of entries in $G_{\mathcal{S}_{\text{in}}^\star}^{(j)}$ lie outside the interval $[\bar{\nabla}_{\text{in},j} - \varepsilon, \bar{\nabla}_{\text{in},j} + \varepsilon]$, hence:

$$\left\|\text{EWM}(G_{\mathcal{S}_{\text{in}}^\star}) - \bar{\nabla}_{\text{in}}\right\|_2^2 \geq \varepsilon^2 d. \tag{47}$$

Now consider a new subset $\widehat{\mathcal{S}}$ formed by swapping the $m_{\text{sep}}^\star$ separated OODs in $\mathcal{S}_{\text{in}}^\star$ with an equal number of InD points whose gradients lie inside $B_\varepsilon$. Such inliers exist because fewer than $m_{\text{out}}\eta_{\text{out}}$ are ambiguous, and the total number of InD points is $m_{\text{in}}$.

Then at least half the entries in every coordinate of $G_{\widehat{\mathcal{S}}}^{(j)}$ lie inside the interval $[\bar{\nabla}_{\text{in},j} - \varepsilon, \bar{\nabla}_{\text{in},j} + \varepsilon]$, implying:

$$\left\|\text{EWM}(G_{\widehat{\mathcal{S}}}) - \bar{\nabla}_{\text{in}}\right\|_2^2 \leq \varepsilon^2 d. \tag{48}$$

Combining with equation 47 yields a contradiction:

$$\left\|\bar{\nabla}_{\text{in}} - \text{EWM}(G_{\widehat{\mathcal{S}}})\right\|_2^2 < \left\|\bar{\nabla}_{\text{in}} - \text{EWM}(G_{\mathcal{S}_{\text{in}}^\star})\right\|_2^2. \tag{49}$$

This contradicts the optimality of $\mathcal{S}_{\text{in}}^\star$, and so the assumption must be false. Therefore, our assumption must be false. With probability at least $1 - \delta$, we conclude that the outlier retention rate is bounded by

$$\text{ERR}_{\text{out}} \leq 2d\exp\left(-\frac{(\Delta - \varepsilon)^2}{2\sigma_{\text{out}}^2}\right) + \sqrt{\frac{\log(1/\delta)}{2m_{\text{out}}}} + \frac{1-\pi}{2\pi}, \tag{50}$$

$\square$

## C.3 THEOREM C.3 WITHOUT SUB-GAUSSIANITY

**Theorem C.3** (Inlier Misclassification Bound without Sub-Gaussianity). *Assume that the gradients of InD points in $\mathcal{S}_{\text{wild}}$ are i.i.d., and that each coordinate has variance $\sigma^2$ and finite fourth moment bounded by $\mu_4$, i.e.,*

$$\mathbb{E}\left[(\nabla\ell(f_\phi(x))_j - \bar{\nabla}_{\text{in},j})^2\right] = \sigma^2, \qquad \mathbb{E}\left[(\nabla\ell(f_\phi(x))_j - \bar{\nabla}_{\text{in},j})^4\right] \leq \mu_4.$$

*Let $m_{\text{in}} = (1-\pi)m$ denote the number of InD samples in $\mathcal{S}_{\text{wild}}$, and fix any tolerance $\varepsilon > \sigma$ and confidence level $\delta \in (0,1)$. Then, with probability at least $1 - \delta$, the inlier misclassification rate of the element-wise median (EWM) filtering rule satisfies:*

$$\text{ERR}_{\text{in}} \leq 2\underbrace{\left(\frac{\mu_4 - \sigma^4}{d(\varepsilon^2 - \sigma^2)^2} + \sqrt{\frac{\log(1/\delta)}{2m_{\text{in}}}}\right)}_{\textit{Concentration term}} + \underbrace{\frac{\pi}{2(1-\pi)}}_{\textit{Contamination term}}.$$

**Differences with the original bound.** The original bound, which assumes sub-Gaussianity, features a decay term

$$2\exp\left(-\frac{\epsilon^2 d}{2\sigma_{\text{sub}}^2}\right)$$

that decreases exponentially with dimension $d$, making the bound extremely tight for large $d$. In contrast, the new bound, relying solely on the assumption of finite fourth moments, exhibits a decay term

$$\frac{\mu_4 - \sigma^4}{d(\epsilon^2 - \sigma^2)^2}$$

that decays as $O(1/d)$, yielding a looser bound for larger $d$. The original bound requires sub-Gaussian tails, while the new one only assumes finite fourth moments, allowing for heavier tails. In particular, both bounds share the finite-sample term

$$\sqrt{\frac{\log(1/\delta)}{2m_{\text{in}}}},$$

that shrinks as $m_{\text{in}}$ increases, independent of the tail assumptions.

*Proof.* The proof follows the structure of the original Theorem 4.1, consisting of two main steps: (1) establishing a concentration bound on the fraction of "bad" InD samples (those far from $\nabla_{\text{in}}$) without relying on sub-Gaussianity, and (2) using a swapping argument to bound the inlier misclassification rate. The key difference is the replacement of the sub-Gaussian tail bound with a moment-based bound derived from the finite fourth moment assumption.

**Step 1: Concentration of InD Gradients** Define the gradient deviation for an InD sample $x$ as:

$$v = \nabla\ell(f_\phi(x)) - \nabla_{\text{in}} \in \mathbb{R}^d, \tag{51}$$

where $\nabla_{\text{in}} = \frac{1}{n}\sum_{(x_i,y_i)\in S_{\text{in}}} \nabla\ell(f_\phi(x_i))$ is the mean gradient over the labeled InD training set $S_{\text{in}}$, and we assume $\nabla_{\text{in}}$ is the population mean. An InD sample is classified as "bad" if:

$$\|v\|_2 > \epsilon\sqrt{d}, \tag{52}$$

and "good" otherwise. Our goal is to bound the fraction of bad inliers among the $m_{\text{in}}$ InD samples in $S_{\text{wild}}$.

Since $\|v\|_2^2 = \sum_{j=1}^d v_j^2$, we analyze the $\ell_2$ norm via the sum of squared coordinates. Assume that the coordinates $v_j = \nabla\ell(f_\phi(x))_j - \nabla_{\text{in},j}$ are independent across $j = 1,\ldots,d$, each with:

$$\mathbb{E}[v_j] = 0 \quad (\text{assuming } \nabla_{\text{in},j} = \mathbb{E}[\nabla\ell(f_\phi(x))_j]), \tag{53}$$

$$\mathbb{E}[v_j^2] = \sigma^2, \quad \mathbb{E}[v_j^4] \le \mu_4. \tag{54}$$

Thus:

$$\mathbb{E}[\|v\|_2^2] = \mathbb{E}\left[\sum_{j=1}^d v_j^2\right] = d\sigma^2, \tag{55}$$

$$\text{Var}(\|v\|_2^2) = \text{Var}\left(\sum_{j=1}^d v_j^2\right) = \sum_{j=1}^d \text{Var}(v_j^2). \tag{56}$$

Compute the variance of $v_j^2$:

$$\text{Var}(v_j^2) = \mathbb{E}[v_j^4] - (\mathbb{E}[v_j^2])^2 \leq \mu_4 - \sigma^4, \tag{57}$$

so:

$$\text{Var}(\|v\|_2^2) \leq d(\mu_4 - \sigma^4). \tag{58}$$

Using Chebyshev's inequality for $\|v\|_2^2$:

$$\mathbb{P}\left(|\|v\|_2^2 - d\sigma^2| \geq t\right) \leq \frac{d(\mu_4 - \sigma^4)}{t^2}. \tag{59}$$

We need $\mathbb{P}(\|v\|_2 > \epsilon\sqrt{d})$, which corresponds to $\|v\|_2^2 > \epsilon^2 d$. Set the threshold:

$$\|v\|_2^2 > \epsilon^2 d \implies \|v\|_2^2 - d\sigma^2 > \epsilon^2 d - d\sigma^2 = d(\epsilon^2 - \sigma^2). \tag{60}$$

Thus:

$$\mathbb{P}(\|v\|_2^2 > \epsilon^2 d) = \mathbb{P}\left(\|v\|_2^2 - d\sigma^2 > d(\epsilon^2 - \sigma^2)\right) \leq \frac{d(\mu_4 - \sigma^4)}{[d(\epsilon^2 - \sigma^2)]^2} = \frac{\mu_4 - \sigma^4}{d(\epsilon^2 - \sigma^2)^2}, \tag{61}$$

provided $\epsilon > \sigma$ so that $\epsilon^2 - \sigma^2 > 0$. Define:

$$p = \mathbb{P}\left(\|\nabla\ell(f_\phi(x)) - \nabla_{\text{in}}\|_2 > \epsilon\sqrt{d}\right) \leq \frac{\mu_4 - \sigma^4}{d(\epsilon^2 - \sigma^2)^2}. \tag{62}$$

Let $X$ be the number of bad inliers among the $m_{\text{in}}$ InD samples in $S_{\text{wild}}$, where each InD sample is bad with probability at most $p$. Then:

$$\mathbb{E}[X] \leq m_{\text{in}}p \leq m_{\text{in}} \cdot \frac{\mu_4 - \sigma^4}{d(\epsilon^2 - \sigma^2)^2}. \tag{63}$$

Since $X = \sum_{i=1}^{m_{\text{in}}} I_i$, where $I_i = 1$ if the $i$-th InD sample is bad and 0 otherwise, and $I_i$ are i.i.d. Bernoulli($p$) variables bounded in $[0, 1]$, apply Hoeffding's inequality:

$$\mathbb{P}\left(X \geq \mathbb{E}[X] + t\right) \leq \exp\left(-\frac{2t^2}{m_{\text{in}}}\right). \tag{64}$$

Set $t = \sqrt{\frac{m_{\text{in}}\log(1/\delta)}{2}}$, so:

$$\mathbb{P}\left(X \geq \mathbb{E}[X] + \sqrt{\frac{m_{\text{in}}\log(1/\delta)}{2}}\right) \leq \delta. \tag{65}$$

With probability at least $1 - \delta$:

$$X \leq m_{\text{in}} \cdot \frac{\mu_4 - \sigma^4}{d(\epsilon^2 - \sigma^2)^2} + \sqrt{\frac{m_{\text{in}} \log(1/\delta)}{2}}. \tag{66}$$

The fraction of bad inliers is:

$$\eta = \frac{X}{m_{\text{in}}} \leq \frac{\mu_4 - \sigma^4}{d(\epsilon^2 - \sigma^2)^2} + \sqrt{\frac{\log(1/\delta)}{2m_{\text{in}}}}. \tag{67}$$

This $\eta$ represents the upper bound on the fraction of InD samples whose gradients deviate from $\nabla_{\text{in}}$ by more than $\epsilon\sqrt{d}$, relying only on finite fourth moments and coordinate independence.

**Step 2: Contradiction via a Swapping Argument**  We demonstrate that if $\mathcal{S}^*$ excludes an excessive number of InD, specifically when

$$\text{ERR}_{\text{in}} > 2\eta + \frac{\pi}{2(1 - \pi)}, \tag{68}$$

then it is possible to construct a new set $\widehat{\mathcal{S}}$ whose EWM is strictly closer to $\bar{\nabla}_{\text{in}}$ than that of $\mathcal{S}^*$, thereby contradicting the optimality of $\mathcal{S}^*$.

Let $m_\varepsilon$ be the number of indices $i$ such that either $x_i$ is OOD or $x_i$ is an InD point whose gradient lies outside the closed $\ell_2$ ball

$$B_\varepsilon = \left\{ g \in \mathbb{R}^d : \|g - \bar{\nabla}_{\text{in}}\|_2 \leq \varepsilon\sqrt{d} \right\}. \tag{69}$$

Let $m_{\text{in}}^*$ be the number of InD points contained in the candidate minimizer $\mathcal{S}^*$.

Assume for contradiction that

$$\text{ERR}_{\text{in}} > 2\eta + \frac{\pi}{2(1 - \pi)}, \tag{70}$$

Define the swap size as

$$m_\varepsilon' := \min \left\{ m_\varepsilon, (\text{ERR}_{\text{in}} - \eta)m_{\text{in}} \right\}. \tag{71}$$

This quantity counts how many good inliers (those inside $B_\varepsilon$) are currently excluded from $\mathcal{S}^*$ and available for swapping in.

Construct a new set $\widehat{\mathcal{S}}$ by removing $m_\varepsilon'$ points from $\mathcal{S}^*$ that lie outside $B_\varepsilon$ (OOD points or bad inliers) and inserting $m_\varepsilon'$ good inliers from $B_\varepsilon$ in their place. This maintains cardinality:

$$|\widehat{\mathcal{S}}| = |\mathcal{S}^*|. $$

Recall that

$$\text{ERR}_{\text{in}} := \frac{m_\varepsilon - m_{\text{in}}^*}{m_{\text{in}}}, \tag{72}$$

which measures the fraction of good inliers excluded from $\mathcal{S}^*$. Rearranging, we obtain:

$$m_\varepsilon - m_{\text{in}}^* = \text{ERR}_{\text{in}} \cdot m_{\text{in}}. \tag{73}$$

Now we know, for contradiction, that

$$\text{ERR}_{\text{in}} > 2\eta + \frac{\pi}{2(1 - \pi)}, \tag{74}$$

Subtracting $\eta$ from both sides and multiplying by $m_{\text{in}}$, we obtain:

$$(\text{ERR}_{\text{in}} - \eta)m_{\text{in}} > \left(\eta + \frac{\pi}{2(1-\pi)}\right)m_{\text{in}}. \tag{75}$$

We know that at most a $\eta$ fraction of inliers are outside $B_\varepsilon$, and the fraction of OOD points among all samples is $\pi$. Therefore, the total number of samples in the dataset that lie outside $B_\varepsilon$ satisfies:

$$m_\varepsilon \leq \eta m_{\text{in}} + \frac{\pi}{1-\pi}m_{\text{in}} = \left(\eta + \frac{\pi}{1-\pi}\right)m_{\text{in}}. \tag{76}$$

It follows that

$$\left(\eta + \frac{\pi}{2(1-\pi)}\right)m_{\text{in}} > \frac{1}{2} \cdot \left(\eta + \frac{\pi}{1-\pi}\right)m_{\text{in}} \geq \frac{m_\varepsilon}{2}. \tag{77}$$

Combining with the previous inequality gives:

$$(\text{ERR}_{\text{in}} - \eta)m_{\text{in}} > \frac{m_\varepsilon}{2}, \tag{78}$$

So $m'_\varepsilon = m_\varepsilon/2 + \eta m_{\text{in}}$, which implies

$$m_\varepsilon - m'_\varepsilon = \frac{m_\varepsilon}{2} - \eta m_{\text{in}}, \tag{79}$$

$$m^*_{\text{in}} + m'_\varepsilon = m^*_{\text{in}} + \frac{m_\varepsilon}{2} + \eta m_{\text{in}}. \tag{80}$$

Because $\eta m_{\text{in}} > 0$, it follows that

$$m^*_{\text{in}} + m'_\varepsilon > m_\varepsilon - m'_\varepsilon, \tag{81}$$

so the points lying inside $B_\varepsilon$ form a strict majority in $\widehat{\mathcal{S}}$.

Fix any coordinate $j \in \{1, \ldots, d\}$. Let $X^{(j)} = \{x_1^{(j)}, \ldots, x_m^{(j)}\}$ be the $j$-th coordinate values of all gradients in $\mathcal{S}^*$ and $\widehat{X}^{(j)} = \{\hat{x}_1^{(j)}, \ldots, \hat{x}_m^{(j)}\}$ the same in $\widehat{\mathcal{S}}$. Let $\mu_j = \bar{\nabla}_{\text{in},j}$.

Each inserted gradient lies inside the interval $I_j = [\mu_j - \varepsilon, \mu_j + \varepsilon]$, and each removed gradient lies outside $I_j$. So for some coordinate $j^*$, the number of values inside $I_{j^*}$ increases from below $m/2$ to above $m/2$ due to the swaps. Therefore:

$$\text{med}(X^{(j^*)}) \notin I_{j^*}, \qquad \text{med}(\widehat{X}^{(j^*)}) \in I_{j^*}, \tag{82}$$

which implies

$$|\text{med}(\widehat{X}^{(j^*)}) - \mu_{j^*}| \leq \varepsilon \quad \text{and} \quad |\text{med}(\widehat{X}^{(j^*)}) - \mu_{j^*}| < |\text{med}(X^{(j^*)}) - \mu_{j^*}|. \tag{83}$$

For the remaining coordinates $j \neq j^*$, the swapped points either leave the majority unchanged or increase it, so the median distance from $\mu_j$ does not increase and possibly decreases.

Thus, component-wise:

$$|\text{EWM}_j(G_{\widehat{\mathcal{S}}}) - \mu_j| \leq |\text{EWM}_j(G_{\mathcal{S}^*}) - \mu_j|, \tag{84}$$

with strict inequality in at least one coordinate $j^*$, implying

$$\|\text{EWM}(G_{\widehat{\mathcal{S}}}) - \bar{\nabla}_{\text{in}}\|_2 < \|\text{EWM}(G_{\mathcal{S}^*}) - \bar{\nabla}_{\text{in}}\|_2. \tag{85}$$

This contradicts the assumption that $\mathcal{S}^*$ is a minimizer of the optimization problem, which concludes the proof.

$$\Rightarrow \quad \text{ERR}_{\text{in}} \leq 2\eta + \frac{\pi}{2(1-\pi)}. \tag{86}$$

$\square$

