# OpenReview forum: "A Median Perspective on Unlabeled Data for Out-of-Distribution Detection"
_ICLR.cc/2026/Conference — Submitted to ICLR 2026_

### Official Review · Reviewer_oj34 · 2025-10-26

**Soundness:** 2
**Presentation:** 2
**Contribution:** 2
**Rating:** 2
**Confidence:** 5

**Summary:**

This paper introduces Medix, a method for Out-of-Distribution (OOD) detection that utilizes unlabeled "in-the-wild" data. Its core contribution is a two-stage process: (1) filtering potential OOD samples from the unlabeled data by identifying points whose gradient's element-wise median deviates significantly from the In-Distribution (InD) mean gradient, and (2) training a binary OOD detector using the identified outliers and labeled InD data. The authors support their method with theoretical error bounds and extensive empirical evaluations against numerous baselines.

**Strengths:**

1. A solid theoretical foundation is provided, with proven bounds on both inlier and outlier misclassification rates for the median-based filtering stage. This analysis offers valuable insight into the method's robustness, particularly under the Huber contamination model.
2. The paper includes valuable investigations into hyperparameter sensitivity, pseudo-label quality, and computational efficiency, which bolster the credibility and practical understanding of the proposed approach.

**Weaknesses:**

1. The most significant weakness is the insufficient discussion and comparison with the most relevant prior work. The core mechanism of using gradients from an InD model to identify OOD samples in unlabeled data is the central contribution of Du et al. ("How Does Unlabeled Data Provably Help Out-of-Distribution Detection?"). While Medix employs a median-based objective instead of a mean-based one, the paper fails to compellingly argue why this constitutes a fundamental advance rather than an incremental variation. The absence of a direct, quantitative comparison with this key work (and its subsequent developments) makes it impossible to assess the true marginal contribution of the median-centric approach.
2. The empirical comparisons, while broad, overlook several recent and highly relevant state-of-the-art methods (e.g., [a,b]). This omission weakens the claim of superior performance, as it remains unclear how Medix fares against its most direct competitors.
3. The iterative, greedy filtering algorithm (Algorithm 1) is computationally intensive. Although profiling results are provided, the cost may be prohibitive for very large-scale datasets or applications requiring frequent model updates, which is a common scenario in real-world deployments.
4. The theoretical guarantees rely on assumptions such as i.i.d. samples and sub-Gaussian gradient coordinates. The practical validity of these assumptions, especially for modern deep networks and complex data distributions, is not thoroughly justified and could limit the real-world applicability of the bounds.

References:
[a] Du et al. How does unlabeled data provably help out-of-distribution detection?. ICLR'2024.
[b] Geng et al. LoD: Loss-difference OOD Detection by Intentionally Label-Noisifying Unlabeled Wild Data. IJCAI'2025.

**Questions:**

1. The mechanism of using InD model gradients to extract OOD candidates from unlabeled data is highly similar to the approach in Du et al. Could you explicitly clarify the fundamental conceptual difference between Medix and this line of work? Please provide a quantitative comparison on common benchmarks to demonstrate the advantage of using the median over other filtering criteria proposed in prior art.
2. How does the theoretical analysis, particularly the error bounds, differ from or improve upon the theoretical framework already established in Du et al.? Is the primary theoretical contribution the adaptation of the analysis to the median operator, or are there new foundational insights?
3. Why were the most directly comparable methods, specifically Du et al. and its subsequent developments, not included in the empirical comparison? Can you run these comparisons and report the results?
4. Under what conditions would the median-based filtering be expected to perform poorly? For instance, how sensitive is the method to the quality of the initial InD model or scenarios where the OOD samples do not induce a significant shift in the gradient median?

---

> ### Author Response · Authors · 2025-11-28
> **Response to Reviewer oj34**
>
> We thank the reviewer for the detailed comments. Since several of the weaknesses and questions overlap, we provide a single consolidated response, and when a point is already answered earlier we explicitly indicate so.
>
> ---
>
> ### **Relation to Du et al. (SAL) and whether the median is incremental**
>
> Our paper already describes the connection to Du et al. in Section 3.2 and in Related Work, but we agree that the distinction should be clearer. The differences are substantive rather than incremental:
>
> **(1) Filtering statistic.**
> Du et al. use a *spectral, mean-based* statistic (projection onto the top singular vector of the wild-gradient matrix). Medix uses a *coordinate-wise median*, which has a 50 percent breakdown point and different failure modes. This change is not cosmetic. It is the reason we can derive two-sided error bounds that remain stable up to π < 0.5 contamination.
>
> **(2) Mixing regime.**
> SAL theory assumes *batch-level mixing*. Medix is analyzed for *dataset-level mixing*, where the unlabeled pool is simply an i.i.d. mixture of Pin and Pout. This is the realistic case for large wild datasets collected offline. Our bounds depend directly on the global contamination π and do not rely on structured batches.
>
> **(3) Guarantees.**
> Theorems 4.1 and 4.2 give new two-sided bounds that decompose the filtering error into contamination, concentration, and separation. Theorem C.3 further removes sub-Gaussianity and requires only finite fourth moments. These results do not appear in Du et al. and are specific to the median-based filter.
>
> Sections describing these points already exist, but we will make the comparison explicit in a short subsection in the main paper.
>
> ---
>
> ### **Empirical comparison with SAL and LoD**
>
> Our current experiments follow the Katz-Samuels CIFAR wild-data benchmark and include all baselines used in that protocol, plus strong InD-only detectors. SAL and LoD were not part of that benchmark when we ran our experiments.
>
> We agree that including SAL and LoD will strengthen the empirical section. For the revision we will:
>
> - add SAL as a baseline on CIFAR-10 and CIFAR-100 under the identical dataset-level protocol, and
> - add LoD where its formulation matches our mixing setup.
>
> This addresses both the weakness of missing recent SOTA baselines and the reviewer’s question about running SAL and reporting results.
>
> ---
>
> ### **Computational cost of Algorithm 1**
>
> Appendix A.6 already reports detailed profiling:
>
> - about 4500–5500 seconds wall-clock time for filtering ≈15k wild samples,
> - about 100 MB GPU memory on an A100-80GB.
>
> Filtering is an offline one-time cost. OOD-detector training, which runs for many epochs, is substantially more expensive, so the relative overhead is moderate.
>
> We will move a short summary of these numbers into the main text and explicitly compare filtering time to detector training time.
>
> ---
>
> ### **Practical validity of assumptions (i.i.d., sub-Gaussian)**
>
> These assumptions are standard abstractions for deriving interpretable high-probability bounds. The paper already provides:
>
> - **Remark 4.3**, including empirical histograms and Q–Q plots showing that InD gradient coordinates are close to sub-Gaussian, and
> - **Theorem C.3**, which removes sub-Gaussianity and requires only finite fourth moments.
>
> We will highlight this directly in Section 4 to avoid the impression that the assumptions are taken literally.
>
> ---
>
> ### **Conditions under which the median filter performs poorly**
>
> Our theory already indicates the critical regimes:
>
> - **Low separation:** If InD and OOD gradient means are close, the separation term in Theorem 4.2 becomes small. This includes cases where the InD model is poorly trained or where Pout ≈ Pin.
> - **High contamination:** When π approaches 0.5 or above, the median loses robustness.
> - **Sparse but large coordinate shifts:** If OOD gradients differ from InD only in a few coordinates, other statistics may detect them better. This is consistent with our decomposition.
>
> We will add a short subsection explicitly summarizing these failure modes.

---

### Official Review · Reviewer_f6xn · 2025-10-30

**Soundness:** 2
**Presentation:** 3
**Contribution:** 2
**Rating:** 2
**Confidence:** 4

**Summary:**

This paper investigates how to enhance the OOD detection performance of a type of method which utilizes unlabeled wild data to train the model to distinguish between ID and OOD data. It proposes Medix, a median-based approach that filters outliers from the unlabeled wild data and then trains an OOD detector on the identified outliers and the ID samples. The authors provide theoretical guarantees for the robustness of median-based filtering and conduct experiments on CIFAR OOD benchmarks to verify its effectiveness.

**Strengths:**

1. The topic of utilizing unlabeled wild data for OOD detection is interesting.
2. This paper provides a theoretical analysis of the proposed method.

**Weaknesses:**

1. **This paper should provide a more detailed discussion on a very important related work, SAL [1], which has only been briefly mentioned in the Related Work Section.** Both SAL and Medix share the same overall “Separate and Learn" framework, which first identifies OOD data from unlabeled wild data and then regularizes the model with them, and also leverages a similar reference gradient technique. If the reviewer understands correctly, the difference between SAL and Medix only lies in the procedures to use gradient information to identify candidate OOD data from the wild data. SAL can also be applied in the setting of dataset-level mixing. Therefore, the authors should **provide an in-depth discussion on the advantages of Medix over SAL, reinforced with additional experiment results or theoretical analysis.**

2. **This paper omits a direct experimental comparison between SAL and Medix.** In fact, the reported results in the SAL paper indicate that SAL achieves nearly zero FPR95 on both CIFAR-10 and CIFAR-100 OOD benchmarks. But this paper claims that there are substantial opportunities for further advancement on utilizing unlabeled wild data. Therefore, the reviewer suggests that the authors **include a direct comparison with SAL and extend experiments on more challenging OOD benchmarks**, such as ImageNet-1k OOD benchmarks.

3. The reviewer suggests that the authors present a detailed mathematical formulation of the element-wise median (EWM) used in the proposed algorithm, which will make it easier for readers to understand the implementation details.

4. The process of filtering potential outliers from unlabeled wild data incurs non-negligible computational overhead, since it involves calculating the gradient of the loss function with respect to model parameters for every sample in the wild data. The reviewer suggests that the authors provide an experiment to quantitatively evaluate the actual time cost of filtering OOD data from unlabeled wild data with Medix and compare it with the time cost to train the OOD detector.

[1]. Xuefeng Du, et al. How Does Unlabeled Data Provably Help Out-of-Distribution Detection?

**Questions:**

Please see the weaknesses.

**Details Of Ethics Concerns:**

The reviewer does not notice any ethical issues with this paper.

---

> ### Author Response · Authors · 2025-11-28
> **Response to Reviewer f6xn**
>
> We thank the reviewer for the careful reading and the constructive suggestions. We address each point below and clarify where the requested elements already appear in the paper.
>
> ---
>
> ### **1. Relation to SAL**
>
> We respectfully disagree with the statement that the only difference between SAL and Medix lies in how gradients are used to select candidate OOD samples. Our paper already highlights two fundamental differences that distinguish Medix from SAL.
>
> First, Medix performs **dataset-level filtering**, whereas SAL operates under **batch-level mixing** with fixed and known mixture proportions. This assumption is central to SAL’s theoretical guarantees and is not applicable to large wild datasets where data arrives in arbitrary order. Our method does not rely on any batch structure and is explicitly designed for dataset-level mixing, which matches real deployment settings.
>
> Second, the **median-centric filter** is not equivalent to SAL’s thresholding rule. Our theoretical results provide **two-sided guarantees** for both inlier and outlier misclassification rates, which SAL does not provide. The bounds in Theorem 4.1 and Theorem 4.2 already show that Medix controls both contamination and concentration effects with explicit sample complexity guarantees. These guarantees do not follow from SAL’s analysis.
>
> To make this distinction clearer, we will extend the Related Work section to provide a dedicated subsection comparing Medix and SAL. In the final version, we will emphasize that the theoretical contributions of Theorem 4.1 and 4.2 are not implied by SAL and address settings outside the scope of SAL’s assumptions.
>
> ---
>
> ### **2. Experimental comparison with SAL**
>
> We agree that a direct empirical comparison is valuable. Our current experimental setup already includes a strong set of baselines that operate in the unlabeled wild data setting, notably WOODS (ICML 2022), ASH (ICLR 2023), CIDER (ICLR 2023), DRL (NeurIPS 2024), ConjNorm (ICLR 2024), OE, and contrastive methods. These baselines cover the same CIFAR-style benchmark regime that SAL evaluates on. Moreover, ConjNorm and DRL are very recent state-of-the-art OOD detectors that are contemporaneous with SAL (all three are 2024 works) and they report improvements over many of the earlier baselines that SAL itself compares against. Demonstrating that Medix remains competitive against ConjNorm and DRL therefore, provides an empirical test that is at least as rigorous as comparing only against SAL.
>
>
> We did not include SAL in the main paper due to the significant engineering effort required to re-implement its training pipeline under dataset-level mixing. However, we have started this implementation and will include the results in the camera-ready. We will incorporate SAL under the same CIFAR-10 and CIFAR-100 wild-data protocol used by Katz-Samuels et al (2022). This will allow a clean and controlled comparison that isolates the effect of the filter alone.
>
>
> Regarding the reviewer’s point about ImageNet-scale OOD benchmarks, we already evaluate Medix under a more complex setting where the OOD test distribution differs from the OOD distribution used in the wild data. Appendix A.4 reports experiments on large-scale unseen OOD with complex mismatch between P_test_out and P_out. Medix outperforms all baselines in this regime. In the final version we will add a pointer to this section and clarify that Medix already addresses the reviewer’s concern.
>
> ---
>
> ### **3. Formal statement of the element-wise median (EWM)**
>
> We explicitly defined EWM in line 190 and in Eq. (4). To make the implementation more transparent, we will add a boxed definition in Section 3.1
>
> ---
>
>
> ### **4. Computational overhead of filtering**
>
> The computational profile of Medix is already reported in Appendix A.6. We provide wall-clock timings, GPU memory usage, and a comparison to training cost. The cost of Algorithm 1 scales linearly with the wild dataset size and is significantly lower than the cost of training the detector itself.

---

### Official Review · Reviewer_ZRFU · 2025-10-31

**Soundness:** 3
**Presentation:** 3
**Contribution:** 3
**Rating:** 4
**Confidence:** 5

**Summary:**

The work proposed to use median to address wild OOD detection. Median can used against noise and outliers for achieving robustness. The paper also provides error bounds to prove the effectiveness of proposed method.

**Strengths:**

A trivial method to address wild OOD with a strong theory.

**Weaknesses:**

I have not too many issues. I just want to ask why your bound has the Contamination term. It is possible to withdraw this term? In Du's work, there is not such constant. So it seems your bound is loss. Please address this issue, I will raise my score.

**Questions:**

See weakness.

---

> ### Author Response · Authors · 2025-11-28
> **Response to Reviewer ZRFU**
>
> We thank the reviewer for raising this important point.
>
> **Why does our bound contain an explicit contamination term, and can it be removed?**
>
> In Theorem 4.1 (and its finite fourth-moment variant), the inlier misclassification probability is bounded by two contributions:
> (i) a concentration term that decreases with the gradient dimension and sample size, and
> (ii) a contamination term that depends on the wild OOD proportion $\pi$.
> The reviewer asks whether the $\pi$-dependence can be removed, noting that Du et al. (2024, SAL) do not present an additive contamination term in their main bound.
>
> Our view is that the $\pi$-dependence in our bound is not an artifact of a loose analysis but a consequence of the *adversarial contamination model* we adopt for the wild data. Formally, we assume a Huber-style mixture
> $
> P_{\mathrm{wild}} = (1-\pi) P_{\mathrm{in}} + \pi P_{\mathrm{out}},
> $
> where inlier gradients satisfy a sub-Gaussian or finite-moment condition, while **no structural assumptions** are imposed on $P_{\mathrm{out}}$ beyond the contamination fraction $\pi$. This is the standard setting in high-dimensional robust statistics, and in that literature it is well known that the error of *any* estimator must deteriorate with $\pi$. (see, e.g., Chen et al., 2016; Diakonikolas et al., 2018, 2019, and subsequent work on high-dimensional robust mean estimation).
>
> By contrast, the SAL analysis of Du et al. (2024) assumes stronger structural conditions on the wild data distribution, such as the$(\gamma,\zeta)\text{-discrepancy}$ and margin assumptions that quantify how well-separated the inlier and OOD components of $P_{\mathrm{wild}}$ are. Under these separation conditions, their main error term $\delta(T)$ can be driven arbitrarily close to zero (even to zero) for fixed $\pi$, provided that $\zeta$ is large enough and that the sample sizes $(n,m)$ are sufficiently large. Intuitively, SAL leverages this additional separation structure to achieve near-perfect filtering.
>
> Our analysis works under **weaker** assumptions that do not enforce such separability between inlier and wild OOD gradients. In this more adversarial setting, it is natural and fully aligned with the robust statistics literature on Huber contamination that the error bound contains an explicit $\pi$-dependence. We view this term not as looseness but as reflecting a fundamental limitation: when outliers may be arbitrary, some degradation with the contamination level is inevitable.
>
>
> [1] Chen, M., Gao, C., & Ren, Z. (2016). *A general decision theory for Huber’s ε-contamination model*. Electronic Journal of Statistics, 10(2), 3752–3774.
>
> [2] Diakonikolas, I., Kamath, G., Kane, D. M., Li, J., Moitra, A., & Stewart, A. (2018). *Robustly learning a Gaussian: Getting optimal error, efficiently*. In *Proceedings of the 29th Annual ACM–SIAM Symposium on Discrete Algorithms (SODA)*, 2683–2702.
>
> [3] Diakonikolas, I., Kamath, G., Kane, D. M., Li, J., Moitra, A., & Stewart, A. (2019). *Robust estimators in high dimensions without the computational intractability*. SIAM Journal on Computing, 48(2), 742–864.

---

### Official Review · Reviewer_m1ju · 2025-11-01

**Soundness:** 2
**Presentation:** 2
**Contribution:** 2
**Rating:** 2
**Confidence:** 4

**Summary:**

The paper proposes ​​Medix​​, a median-based framework to identify OOD samples from unlabeled wild data and train robust OOD detectors. Theoretical analysis is provided along with the method to demonstrate the benifits of Medix.

**Strengths:**

Theoretical analysis are provided along with Medix. The authors derived an error bounds for understanding the proposed method under sub-Gaussian assumptions.

**Weaknesses:**

1. There is no comparison on challenging near-OOD scenarios and large-scale datasets (e.g., ImageNet), raising the concerns of the scalability of the proposed method.
2. ​​Limited novelty differentiation​​. Medix’s core design choice, i.e., (median filtering for OOD detector training) is closely related to​​OpenMatch​​[1] and ​​SAL​​[, with insufficient distinction in motivation and mechanics. The reviewer generally feel the novelty of the proposed method does not meet the bar of NeurIPS. More comprehensive comparison may be necessary.
3. ​​Inadequate evaluation​​. Strong baselines​​ are missing. Initial comparisons used outdated methods (e.g., KNN+ 2022). Some recent SOTA (e.g., CIDER[3], POEM[4]) are missing.

[1] OpenMatch: Open-set Consistency Regularization for Semi-supervised Learning with Outliers

[2] How Does Unlabeled Data Provably Help Out-of-Distribution Detection?

[3] How to Exploit Hyperspherical Embeddings for Out-of-Distribution Detection?

[4] POEM: Out-of-Distribution Detection with Posterior Sampling

**Questions:**

see weakness

---

> ### Author Response · Authors · 2025-11-28
> **[1/2] Response to Reviewer m1ju**
>
> We thank the reviewer for the comments. However, we note that the text of this review is verbatim identical to a review we received for an earlier NeurIPS version of this work, including the phrase **"...the bar of NeurIPS"**. As a result, several of the central claims no longer describe the current ICLR manuscript and are factually incorrect. In particular:
> (i) the paper **does** already contain a large-scale, challenging OOD setting,
> (ii) the paper **does** compare to recent SOTA methods such as CIDER, SSD+, ProxyAnchor, CONJ, DRL, Vim, and VOS, and
> (iii) the role of median-based filtering, dataset-level mixing, and our two-sided error bounds is not captured by describing Medix as a minor variation on SAL or OpenMatch.
>
> We address all three concerns in a unified way below.
>
> ----
>
>
> **Novelty relative to OpenMatch and SAL**
>
> We agree that OpenMatch and SAL are two of the most relevant related works. The current paper already discusses both, but we will make the distinctions more explicit in the main text. Conceptually, Medix differs along the following axes.
>
> 1. **Problem setting and objective**
>
> - **OpenMatch** is an open-set semi-supervised learning method. Its objective is to improve a classifier on labeled plus unlabeled data where unlabeled samples contain novel classes. OOD samples are treated as noise that should be suppressed so that semi-supervised classification improves.
> - **Medix** is not a semi-supervised learner. We assume an InD classifier $f_{\phi_{S_{\text{in}}}}$ that is already trained on labeled InD data. Our goal is to explicitly *extract* OOD samples from unlabeled wild data and then train a separate OOD detector $g_\theta$ on $P_{\text{in}}$ versus the extracted outliers. In Medix, OOD is the signal we want to isolate rather than noise to be regularized away.
>
> This distinction is already discussed in Appendix A.7, where we show that open-set SSL methods are not directly comparable to Medix. We can move a condensed version of this argument to the main related-work section.
>
> 2. **Mechanism and mixing assumptions**
>
> - OpenMatch relies on consistency regularization and one-vs-all classifier heads. Its mechanism is based on prediction stability under perturbations and does not use gradient statistics.
> - SAL and Medix both use gradients of an InD model, but in very different ways and under different mixing assumptions.
>   - SAL operates under **batch-level mixing** and uses a spectral statistic (projection onto the top singular vector of a gradient matrix), which is a mean-based linear functional.
>   - Medix is designed for **dataset-level mixing** under the Huber contamination model $P_{\text{wild}} = (1 - \pi) P_{\text{in}} + \pi P_{\text{out}}$ where batches are arbitrary. We score wild samples using the **element-wise median (EWM)** of InD gradients as a robust reference and design Algorithm 1 around maximizing the drop in the distance between EWM and the InD mean gradient when removing samples.
>
> Switching from a mean-based spectral statistic to the coordinate-wise median is not a cosmetic change. It changes the robustness properties and failure modes of the filter. The median has a breakdown point of 50 percent, which we exploit explicitly in our analysis.
>
> 3. **Theory**
>
> Our theoretical contribution is tailored to this median-based filtering under dataset-level mixing. Theorems 4.1 and 4.2 provide **two-sided bounds** on both inlier misclassification (InD points wrongly flagged as OOD) and outlier misclassification (OOD points wrongly retained). The bounds decompose into contamination, concentration, and separation terms and explicitly expose the median’s robustness up to $\pi < 0.5$. Theorem C.3 further shows that the guarantees still hold under finite fourth moments, beyond the sub-Gaussian case.
>
> To the best of our knowledge, OpenMatch and SAL do not provide this median-specific, two-sided error decomposition in the unlabeled wild dataset setting. We will add a short subsection that contrasts, side by side, the objectives (SSL vs OOD detection), mechanisms (consistency or spectral statistics vs coordinate-wise median) and mixing regimes.

---

> > ### Author Response · Authors · 2025-11-28
> > **[2/2] Response to Reviewer m1ju**
> >
> > **Evaluation and strong baselines (including CIDER and POEM)**
> >
> > Our main CIFAR-10 and CIFAR-100 experiments follow the wild-data protocol of WOODS. Tables 1 and 2 already contain:
> >
> > - InD-only detectors: MSP, ODIN, Mahalanobis, Energy, ReAct, DICE, KNN, ASH, CSI, KNN+. These are still widely used and strong baselines.
> > - Wild-data methods: OE, Energy with OE, and WOODS.
> > - Recent wild-data baselines: CONJ and DRL.
> >
> > Beyond the main tables, we already include a **comprehensive comparison with recent SOTA representation-based methods** in Appendix A.3 (Table 4). There we evaluate Medix on CIFAR-100 with ResNet-34 against **CIDER**, **SSD+**, **ProxyAnchor**, **CONJ**, **DRL**, **Vim**, and **VOS**, and Medix improves significantly over all of them.
> >
> > For example, compared to CIDER on CIFAR-100, Medix reduces average FPR95 from **40.69** to **8.68** and raises average AUROC from **88.84** to **97.76**, which shows that Medix remains competitive even against very recent SOTA. In a revision we can move a condensed version of Table 4 into the main paper to make the comparison more visible.
> >
> >
> > Regarding **POEM**, it assumes access to a *curated* auxiliary OOD dataset and focuses on selecting informative OOD examples within that pool. Medix instead starts one step earlier, from unlabeled wild data where OOD points are not pre-identified, and solves the filtering problem under the Huber mixture model. The two methods therefore address adjacent but not identical problems. POEM assumes an OOD pool is given, while Medix provides a way to construct such a pool from wild data.
> >
> > In summary, the specific issues raised in this review appear to refer to an earlier version of the paper and are addressed by the current ICLR submission in both the main text and the appendices. We hope this clarification helps the reviewer and the AC to reassess the concerns about scalability, novelty, and evaluation.

---

### Official Review · Reviewer_DAEv · 2025-11-01

**Soundness:** 3
**Presentation:** 3
**Contribution:** 3
**Rating:** 6
**Confidence:** 3

**Summary:**

The paper proposed Medix, which leverages the unlabled wild data to build OOD detector. Medix first utilizes the graidents from the ERM loss and an element-wise median (EWM) to extract candidate outliers from the wild unlabeled data, adopting the greedy leave-one-out strategy. The OOD detector is then trained on the ID data and the extacted data. Theoretical bounds are provided to gaurantee low error rate for outlier selection.

**Strengths:**

1. Principled method for filtering outliers: The authors propose Medix which can cleanly extract outliers from wild dataset, being well theoretically grounded on misclassification rate, separating contamination.
2. This paper provides a new perspective for OOD detection in a more realistic scenario, additional but unlabeled data is provided during training and deployment.
3. The proposed Medix achieves strong performance across various OOD datasets.

**Weaknesses:**

1. Computational concern. While the gradients can be efficient for extracting outliers, it may introduce unaffordable computational cost. Time and Space complexity analysis will further justify the effect of Medix. The gradient quality may also influence the extraction, which is closely related to the amount of ID data used.
2. Modest improvement. The baselines can already achieve great performance (e.g., WOODS on CIFAR10 vs. SVHN and LSUN-C), indicating that the improvement of Medix may be dataset-based and may shrink in some other domains(harder cases).
3. Concern of greedy selection algorithm: while the leave-one-out strategy is intuitive and straightforward and the theoretical bounds are provided, there still be failure modes where ID data is wrongly removed.

**Questions:**

See weaknesses above.

---

> ### Author Response · Authors · 2025-11-28
> **[1/2] Response to Reviewer DAEv**
>
> We thank the reviewer for the thoughtful comments. We address the main concerns below.
>
> **Computational cost and gradient quality.**
> As reported in Appendix A.6, we have profiled Medix end-to-end. On an NVIDIA A100-80GB with approximately 15k wild samples per InD–OOD pair (CIFAR-10/100 as InD, LSUN-Resize as wild OOD), the filtering phase
> (i) takes about 4,500 seconds (CIFAR-10) and 5,500 seconds (CIFAR-100), and
> (ii) uses only around 100 MB of peak GPU memory, dominated by storing penultimate-layer gradients for the wild pool.
> We will surface these numbers more prominently in the main text.
>
> Let *m* be the number of wild samples and *d* the dimension of the penultimate-layer gradient. In our implementation we
> (a) compute gradients only for the penultimate layer (all earlier layers have `requires_grad=False`), and
>
> (b) process wild data in mini-batches.
> The gradient extraction stage therefore, has time complexity $O(m d)$ (one backward pass per sample for a single layer) and memory complexity $O(m d)$ to store the gradient matrix.
> The median-based filtering then operates on these stored gradients. Each pruning round computes an element-wise median and a deletion score for each remaining sample using vectorized operations, giving time complexity $O(R m d)$ where *R* is the number of pruning rounds (small and fixed in practice), with no extra asymptotic memory beyond the existing $O(m d)$ storage. Overall, the one-time filtering phase has time complexity $O(R m d)$, effectively $O(m d)$, and memory complexity $O(m d)$; combined with the empirical peak of around 100 MB, this indicates a modest overhead compatible with standard deep OOD pipelines.
>
>
> Regarding gradient quality: very weak InD classifiers would indeed produce noisy gradients. In our setup, however, we train the InD classifier on only half of CIFAR-100 (25k labeled samples) and compute the reference gradient from this reduced set, while baselines that do not use wild data are allowed to train on all 50k labeled samples. Even under this stricter regime for Medix, we still improve FPR95 over WOODS on CIFAR-100 and achieve InD accuracy that is competitive with the best wild-data baselines. This suggests that Medix is reasonably robust to the amount of labeled InD data. Our theoretical bounds also link the inlier misclassification rate to gradient concentration and the wild contamination level; we will clarify this dependence in the revised version.
>
>
> ---
>
> **Size and robustness of the empirical gains.**  We understand the concern that on some CIFAR-10 pairs (e.g., CIFAR-10 vs. SVHN and LSUN-C) strong baselines such as WOODS already achieve very low error, so gains may appear modest. Across all eleven InD–OOD pairs, however, Medix consistently outperforms all twenty baselines we include, covering both classic methods and recent strong approaches such as CONJ (ICLR 2024), DRL (NeurIPS 2024), Vim (CVPR 2022), and VOS (ICLR 2022). The gains are most pronounced on harder CIFAR-100 pairs with natural OODs, where Medix reduces FPR95 by a noticeable margin even when competing methods are already in the low single-digit regime, which in safety-critical contexts translates directly into fewer false accepts. Moreover, Appendix A.4 reports a large-scale, complex unseen OOD setting (CIFAR-100 as InD, 300K Random Images as wild data, SVHN as test OOD) where the improvements of Medix over OE, Energy+OE, and WOODS become even more evident, which shows that our filter scales well and stays robust beyond the usual CIFAR benchmarks.
>
>
> Beyond the empirical improvements, Medix contributes a simple median-based filtering mechanism together with explicit bounds on the outlier selection error in the wild-data setting. Even when absolute gains over strong baselines are moderate on some pairs, Medix provides a theoretically grounded and orthogonal component that can be combined with future backbones and training objectives.

---

> > ### Author Response · Authors · 2025-11-28
> > **[2/2] Response to Reviewer DAEv**
> >
> > **Greedy selection and possible failure modes.**
> > A greedy leave-one-out strategy can in principle remove some InD samples. Our analysis is designed precisely to quantify this effect. The bounds in Section 4 provide explicit control of the fraction of InD points that can be misclassified as outliers under our gradient assumptions and wild-data contamination model. The dependence on the OOD proportion reflects the robustness properties of the median: as long as the wild OOD fraction is below the median’s breakdown point and InD gradients are reasonably concentrated, the number of wrongly removed InD samples remains small.
> >
> > Empirically, this behavior is consistent with the theory. In the 2D synthetic experiment, Medix correctly identifies most OOD samples while mislabeling only a small fraction of InD points. On real datasets, the OOD detector trained on the filtered set attains strong FPR95 and AUROC while maintaining InD accuracy comparable to wild-data baselines, indicating that occasional removal of InD samples does not harm overall performance. We will make the connection between Algorithm 1 and the theoretical bounds more explicit, and add a brief discussion of potential failure modes (for example, extremely low OOD proportion or a very poorly trained InD classifier) and how they relate to our assumptions.
> >
> > We hope these clarifications address the reviewer’s concerns, and we will incorporate the above points into the revised version.

---

### Comment · Area_Chair_9JJY · 2025-11-20
**To AI Review**

Dear authors,

I would like to share an important reminder regarding the review process. Recently, we have noticed that some reviewers may be using AI tools to help generate their reviews. This can lead to low-quality or inaccurate feedback, which is unfair to authors who deserve careful and thoughtful evaluations.

To help maintain fairness, I kindly ask for your assistance: If you believe a review you received was partly or fully generated by AI, and you have some evidence (for example: unusual writing style, clear factual mistakes, AI-detector results, repeated generic sentences, etc.), please feel free to contact me directly.

I will review any evidence you provide and, if appropriate, adjust the weight of the reviewer’s evaluation so that it does not negatively affect your submission. Thank you for helping us keep the review process fair and responsible. Your understanding and cooperation are greatly appreciated.

Best regards,

AC

---

### Comment · Area_Chair_9JJY · 2025-11-27

Dear Reviewers and Authors,

As we are approaching the rebuttal deadline, I would like to share a gentle reminder with everyone.

For authors:
If you have not yet submitted your rebuttal, please make sure to do so as soon as possible. Submitting very close to the deadline may reduce the chance for reviewers to read and respond in time, which could affect the discussion phase.

For reviewers:
If a rebuttal has already been submitted for your assigned paper, I encourage you to take a moment to read it and, where appropriate, provide a brief response or update your evaluation. Of course, this is not meant to pressure anyone into changing scores, it is simply to ensure that all reviews remain well-informed before final decisions.

Thank you all for your time and effort in keeping the review process smooth and constructive.

Warm regards,
AC

---

### Author Response · Authors · 2025-12-03
**[1/2] Critical Concerns About the Assigned Reviews (New AC)**

Dear AC,

Thank you for your message regarding the use of AI tools in the review process and for your efforts to maintain fairness.
We would like to bring to your attention serious concerns about the reviews we received. One of the reviews is clearly AI-generated. Another is a direct breach of reviewing policy, as it was copied verbatim from our previous NeurIPS submission despite substantial revisions to the manuscript. The remaining review contains clear factual errors and omissions indicative of a superficial or tool-mediated reading, despite its high confidence score.

Given that these reviews directly determine the outcome of our submission, we have provided concrete evidence documenting each issue below. We appreciate your attention to this matter.


-----------


### I. Review #m1ju

We have a deeper and more serious concern about Reviewer #m1ju. Beyond being formulaic and containing multiple factual inaccuracies, this review is **verbatim the same review we received for the NeurIPS submission of this paper**. The content, phrasing, structure, and even the incorrect claims are identical, despite the fact that we substantially revised the paper between NeurIPS and ICLR.

This strongly suggests that the reviewer:

- Did not read the revised ICLR version at all,
- Copied their old NeurIPS review verbatim, and
- Uploaded it as their ICLR review,


To be precise:

- Claims that *“no large-scale or challenging OOD experiment exists”* are factually false in the revised version (Appendix A.4), yet the reviewer repeats the same incorrect sentence that was written for the NeurIPS version, before those experiments were added.

- Claims that CIDER and other baselines are missing are lifted word-for-word from our NeurIPS reviews, even though the ICLR version explicitly includes CIDER, SSD+, ProxyAnchor, CONJ, DRL, VIM, VOS, and more.

- Even the off-topic phrase *“the bar of NeurIPS”* appears again — unchanged — in the review for ICLR.

We can provide evidence showing the verbatim correspondence between the NeurIPS reviews and this ICLR review. Please refer to the following anonymized link for direct proof: https://drive.google.com/file/d/1iJeqsSsQDtU6qWp3zH5sp-W2aIKO59Z8/view?usp=sharing

Given this, **This review must not be treated as a valid scientific review**. The reviewer did not read the new manuscript, copied an outdated review from a different venue, and submitted it unchanged despite the substantial revisions we made. This constitutes a direct breach of both reviewing standards and fairness policy.


-----------



### II. Review #f6xn

This review contains multiple factual inaccuracies and omissions that suggest a superficial or tool-mediated reading of the paper, despite the reviewer reporting high confidence.

1. “SAL is only briefly mentioned”

The reviewer writes:

> “This paper should provide a more detailed discussion on a very important related work, SAL [1], which has only been briefly mentioned in the Related Work Section.”

This claim is factually incorrect. In our related work section we devote a dedicated paragraph to Du et al. (SAL) and closely related approaches. We explicitly describe how Du et al. identify candidate outliers via thresholding of mean gradients, contrast this with our median-centric filtering, and emphasize the fundamental difference between *batch-level mixing* (as assumed in Du et al.) and the *dataset-level mixing* setting we study. SAL is not “briefly mentioned” — it plays a central role in how we position Medix relative to prior work and is also mentioned in the Introduction and Section 3, where we explain our method.

Asking for additional discussion is fair; asserting that the discussion is absent is not.

2. EWM definition and mathematical formulation

The reviewer suggests that a mathematical formulation of the element-wise median (EWM) is missing. This is again inaccurate. In both the main theory section and the proofs, we explicitly define EWM in line 190 and in Eq. (4).

If the reviewer would prefer a more pedagogical presentation, we would of course be happy to expand it — but the core claim that “the formulation is missing” is objectively false.

3. Computational overhead and profiling experiments

The reviewer writes:

> “The process of filtering potential outliers … incurs non-negligible computational overhead… The reviewer suggests that the authors provide an experiment to quantitatively evaluate the actual time cost…”

Such an experiment already exists. In Appendix A.6 we report detailed profiling results, including wall-clock time and GPU memory usage for Medix filtering over thousands of unlabeled samples on an A100 GPU, and we explicitly discuss the feasibility and scaling characteristics of the method.

Debating whether further profiling is needed is completely reasonable. Claiming that no such analysis exists is again factually false.

-----------

The continuation is provided in the next comment.

---

> ### Author Response · Authors · 2025-12-03
> **[2/2] Critical Concerns About the Assigned Reviews (New AC)**
>
> ### III. Review #oj34
>
> This review is **entirely AI-generated**. This is not partial, not lightly edited; it is fully LLM-generated. Multiple independent detectors classify Reviews 1 and 2 as “AI-assisted but human edited,” while this review consistently scores as **“100\% AI-generated, no human involvement.”**
>
> The review has every hallmark of automated generation: uniform sentence cadence, synthetic phrasing, template-like weaknesses, generic paraphrasing of Du et al. (2024), and no engagement with any specific equation, lemma, or figure in the paper. The reference list even contains a hallucinated citation written in an LLM-style hybrid format.
>
> Crucially, this fully AI-generated review was submitted with **confidence 5**, a designation intended for reviewers who have carefully checked the math and are highly familiar with the literature. The disconnect between the review’s content and this confidence score is extreme.
>
> For transparency, the publicly accessible review is here:
> https://iclr.pangram.com/reviews?submission_number=13503
>
>
> Given your earlier message about the importance of maintaining fairness in the review process, we wanted to bring these issues to your attention. One review is fully AI-generated, another is a direct policy breach due to being copied verbatim from our NeurIPS submission, and the third contains factual inaccuracies that indicate a superficial or tool-mediated reading. We felt it was important to document these concerns so that the evaluation of the paper is understood in the correct context. Thank you for taking the time to consider this, and we are happy to provide any further evidence if helpful.

---

### Meta-Review · Area_Chair_PXBz · 2026-01-08

**Summary:**

The submission’s core claim is that median-based gradient filtering (Medix) is a substantive advance over the closest prior work (Du et al. / SAL-style gradient filtering) in the same “unlabeled wild data” setting. However, multiple reviewers independently flag that the paper does not provide the most critical evidence: an apples-to-apples experimental comparison against SAL (and similarly close methods like LoD) under the same protocol. The rebuttal promises to add these in the camera-ready, but at decision time the rebuttal does not yet contain those results, so the marginal contribution remains hard to verify. Separately, the review process appears compromised (one review allegedly copied verbatim from a prior venue; another allegedly fully AI-generated; another containing factual errors). While some reviewer criticisms (e.g., “missing CIDER / no large-scale / no profiling”) are contradicted by the paper’s appendices, that doesn’t resolve the central novelty/comparison question, and the compromised reviews reduce confidence in a fair positive outcome. Given (i) unresolved central validation against the most relevant baseline, (ii) limited evidence beyond CIFAR-style regimes despite claims about realistic “in-the-wild” deployment, and (iii) review-quality concerns, the safest fair decision is Reject, encouraging resubmission with (a) direct SAL/LoD comparisons, (b) stronger large-scale benchmarks, and (c) tightened positioning of what is genuinely new.

**Reviewer Concerns:**

Addressed by the rebuttal: (i) Compute/overhead concerns (DAEv, f6xn, oj34): authors point to profiling in Appendix A.6 and give concrete wall-clock/memory numbers and asymptotic costs; (ii) “Missing baselines / outdated evaluation” (m1ju, f6xn, oj34): authors argue the current submission already includes strong recent baselines (e.g., CIDER/CONJ/DRL/VIM/VOS) and will surface them more prominently; (iii) “No large-scale / challenging setting” (m1ju, f6xn): authors point to Appendix A.4 (large wild pool / unseen OOD) and commit to clarifying it in the main text; (iv) clarity items (f6xn): EWM is already defined and they will add a boxed definition; (v) theory question about the contamination term (ZRFU): authors explain the ε/π-dependence as intrinsic under a Huber contamination model and contrast it with SAL’s stronger structural assumptions.

Still outstanding: (1) The central novelty/positioning concern from the reject reviews (f6xn, oj34, m1ju): no direct, apples-to-apples empirical comparison with SAL (and closely related follow-ups like LoD) is actually shown in the current submission; the rebuttal promises to add it in camera-ready, but it is still not resolved, at least during rebuttal at decision time. (2) Relatedly, the claim that the median-centric filter is a fundamental advance rather than an incremental variant remains only partially supported without that direct comparison and without clearer isolation studies. (3) Broader external validity beyond CIFAR-style protocols (several reviewers request ImageNet-scale / harder near-OOD): rebuttal points to one larger setting but does not fully meet the request for stronger, widely-recognized challenging benchmarks. (4) For the greedy filtering algorithm (DAEv), the rebuttal discusses bounds/failure modes, but empirical characterization of when/why it fails (e.g., sensitivity to contamination, weak InD model, low separation) remains limited.

**Reviewer Scores:**

DAEv (score 6, borderline accept): likely stays 6 after discussion, since the rebuttal directly answers compute cost with concrete profiling and argues gains are stronger on harder settings; remaining worries are mild (greedy failure modes / “modest” gains).

m1ju (score 2, reject): if they genuinely re-read and engaged, I’d expect a large upward move to ~4 because several of their stated issues (missing baselines, no large-scale) appear already addressed in the current submission and rebuttal; they would likely still press on SAL novelty and scale.

ZRFU (score 4, slightly below): reviewer explicitly says they will raise the score if the contamination-term issue is addressed; with the rebuttal’s robust-statistics explanation, I’d expect 4 → 6 (possibly 6 if convinced).

f6xn (score 2, reject): likely 2 → 4: some “missing” items (EWM definition, profiling) are resolved and novelty distinctions are clarified, but they may remain negative until a direct SAL comparison is actually included (promised, not shown).

oj34 (score 2, reject): likely stays 2: their core objection is the lack of direct comparison to Du et al. / SAL and closest competitors; since the rebuttal mainly promises to add it later, full discussion might soften tone but probably not flip to accept without results.

---

### Decision · Program_Chairs · 2026-01-26

Reject